# CAUSAL BALANCING FOR DOMAIN GENERALIZATION

**Xinyi Wang**[1], **Michael Saxon**[1], **Jiachen Li**[1], **Hongyang Zhang**[2], **Kun Zhang**[3,4],
**William Yang Wang**[1]

[1]Department of Computer Science, University of California, Santa Barbara, USA
[2]David R. Cheriton School of Computer Science, University of Waterloo, Canada
[3]Department of Philosophy, Carnegie Mellon University, USA
[4]Machine Learning Department, Mohamed bin Zayed University of Artificial Intelligence, UAE
`xinyi_wang@ucsb.edu, saxon@ucsb.edu, jiachen_li@ucsb.edu,`
`hongyang.zhang@uwaterloo.ca, kunz1@cmu.edu, william@cs.ucsb.edu`

## ABSTRACT

While machine learning models rapidly advance the state-of-the-art on various real-world tasks, out-of-domain (OOD) generalization remains a challenging problem given the vulnerability of these models to spurious correlations. We propose a balanced mini-batch sampling strategy to transform a biased data distribution into a spurious-free balanced distribution, based on the invariance of the underlying causal mechanisms for the data generation process. We argue that the Bayes optimal classifiers trained on such balanced distribution are minimax optimal across a diverse enough environment space. We also provide an identifiability guarantee of the latent variable model of the proposed data generation process, when utilizing enough train environments. Experiments are conducted on DomainBed, demonstrating empirically that our method obtains the best performance across 20 baselines reported on the benchmark. [1]

## 1 INTRODUCTION

Machine learning is achieving tremendous success in many fields with useful real-world applications (Silver et al., 2016; Devlin et al., 2019; Jumper et al., 2021). While machine learning models can perform well on in-domain data sampled from seen environments, they often fail to generalize to out-of-domain (OOD) data sampled from unseen environments (Quiñonero-Candela et al., 2009; Szegedy et al., 2014). One explanation is that machine learning models are prone to learning spurious correlations that change between environments. For example, in image classification, instead of relying on the object of interest, machine learning models easily rely on surface-level textures (Jo & Bengio, 2017; Geirhos et al., 2019) or background environments (Beery et al., 2018; Zhang et al., 2020). This vulnerability to changes in environments can cause serious problems for machine learning systems deployed in the real world, calling into question their reliability over time.

Various methods have been proposed to improve the OOD generalizability by considering the invariance of causal features or the underlying causal mechanism (Pearl, 2009) through which data is generated. Such methods often aim to find invariant data representations using new loss function designs that incorporate some invariance conditions across different domains into the training process (Arjovsky et al., 2020; Mahajan et al., 2021; Liu et al., 2021a; Lu et al., 2022; Wald et al., 2021). Unfortunately, these approaches have to contend with trade-offs between weak linear models or approaches without theoretical guarantees (Arjovsky et al., 2020; Wald et al., 2021), and empirical studies have shown their utility in the real world to be questionable (Gulrajani & Lopez-Paz, 2020).

In this paper, we consider the setting that multiple train domains/environments are available. We theoretically show that the Bayes optimal classifier trained on a balanced (spurious-free) distribution is minimax optimal across all environments. Then we propose a principled two-step method to sample balanced mini-batches from such balanced distribution: (1) learn the observed data distribution using a variational autoencoder (VAE) and identify the *latent covariate*; (2) match train examples

---

[1]We publicly release our code at `https://github.com/WANGXinyiLinda/causal-balancing-for-domain-generalization`.

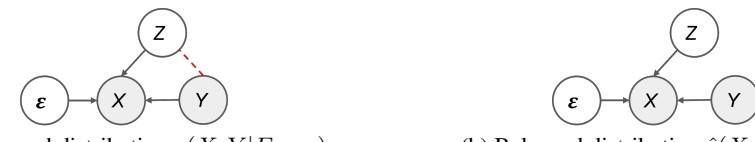

(a) Observed distribution $p(X, Y|E = e)$       (b) Balanced distribution $\hat{p}(X, Y|E = e)$

Figure 1: The causal graphical model assumed for data generation process in environment $e \in \mathcal{E}$. Shaded nodes mean being observed and white nodes mean not being observed. Black arrows mean causal relations invariant across different environments. The Red dashed line means correlation varies across different environments.

with the closest latent covariate to create *balanced mini-batches*. By only modifying the mini-batch sampling strategy, our method is lightweight and highly flexible, enabling seamless incorporation with complex classification models or improvement upon other domain generalization methods.

Our contributions are as follows: (1) We propose a general non-linear causality-based framework for the domain generalization problem of classification tasks; (2) We prove that a spurious-free *balanced distribution* can produce minimax optimal classifiers for OOD generalization; (3) We rigorously demonstrate that the source of spurious correlation, as a latent variable, can be identified given a large enough set of training environments in a nonlinear setting; (4) We propose a novel and principled balanced mini-batch sampling algorithm that, in an ideal scenario, can remove the spurious correlations in the observed data distribution; (5) Our empirical results show that our method obtains significant performance gain compared to 20 baselines on DomainBed (Arjovsky et al., 2020).

## 2 PRELIMINARIES

**Problem Setting.** We consider a standard domain generalization setting with a potentially high-dimensional variable $X$ (e.g. an image), a label variable $Y$ and a discrete environment (or domain) variable $E$ in the sample spaces $\mathcal{X}, \mathcal{Y}, \mathcal{E}$, respectively. Here we focus on the classification problems with $\mathcal{Y} = \{1, 2, ..., m\}$ and $\mathcal{X} \subseteq \mathbb{R}^d$. We assume that the training data are collected from a finite subset of training environments $\mathcal{E}_{\text{train}} \subset \mathcal{E}$. The training data $\mathcal{D}^e = \{(x_i^e, y_i^e)\}_{i=1}^{N^e}$ is then sampled from the distribution $p^e(X, Y) = p(X, Y|E = e)$ for all $e \in \mathcal{E}_{\text{train}}$. Our goal is to learn a classifier $C_\psi : \mathcal{X} \to \mathcal{Y}$ that performs well in a new, unseen environment $e_{test} \notin \mathcal{E}_{\text{train}}$.

We assume that there is a data generation process of the observed data distribution $p^e(X, Y)$ represented by an underlying structural causal model (SCM) shown in Figure 1a. More specifically, we assume that $X$ is caused by label $Y$, an unobserved latent variable $Z$ (with sample space $\mathcal{Z} \in \mathbb{R}^n$) and an independent noise variable $\epsilon$ with the following formulation:

$$X = \mathbf{f}(Y, Z) + \epsilon = \mathbf{f}_Y(Z) + \epsilon.$$

Here, we assume the causal mechanism is invariant across all environments $e \in \mathcal{E}$ and we further characterize $\mathbf{f}$ with the following assumption:

**Assumption 2.1.** $\mathbf{f} : \{1, 2, ..., m\} \times \mathcal{Z} \to \mathcal{X}$ *is injective.* $\mathbf{f}^{-1} : \mathcal{X} \to \{1, 2, ..., m\} \times \mathcal{Z}$ *is the left inverse of* $\mathbf{f}$.

Note that this assumption forces the generation process of $X$ to consider both $Z$ and $Y$ instead of only one of them. Suppose $\epsilon$ has a known probability density function $p_\epsilon > 0$. Then we have

$$p_{\mathbf{f}}(X|Z, Y) = p_\epsilon(X - \mathbf{f}_Y(Z)).$$

While the causal mechanism is invariant across environments, we assume that the correlation between label $Y$ and latent $Z$ is environment-variant and $Z$ should exclude $Y$ information. i.e., $Y$ cannot be recovered as a function of $Z$. If $Y$ is a function of $Z$, the generation process of $X$ can completely ignore $Y$ and $f$ would not be injective. We consider the following family of distributions:

$$\mathcal{F} = \left\{\, p^e(X, Y, Z) = p_{\mathbf{f}}(X \mid Z, Y)p^e(Z|Y)p^e(Y)|p^e(Z|Y), p^e(Y) > 0 \,\right\}_e. \tag{1}$$

Then the environment space we consider would be all the index of $\mathcal{F}$: $\mathcal{E} = \{\, e \mid p^e \in \mathcal{F} \,\}$. Note that any mixture of distributions from $\mathcal{F}$ would also be a member of $\mathcal{F}$. i.e. Any combination of the environments from $\mathcal{E}$ would also be an environment in $\mathcal{E}$.

To better understand our setting, consider the following example: an image $X$ of an object in class $Y$ has an appearance driven by the fundamental shared properties of $Y$ as well as other meaningful latent features $Z$ that do not determine "$Y$-ness", but can be spuriously correlated with $Y$. In

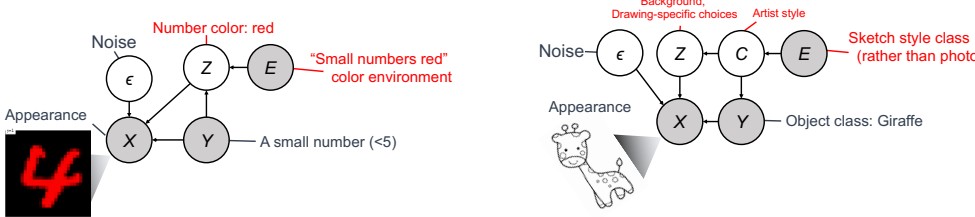

(a) As realized by ColoredMNIST             (b) As realized by PACS

Figure 2: Annotated example causal graphs of two realizations of the joint distribution $p(X, Y, E)$.

Figure 2, we plot causal diagrams for the joint distributions $p(X, Y, E)$ of two example domain generalization datasets, ColoredMNIST (Arjovsky et al., 2020) and PACS (Li et al., 2017). In ColoredMNIST, $Z$ indicates the assigned color, which is determined by the digit label $Y$ and the environment $E = p(Z|Y)$. In PACS, images of the same objects in different styles (e.g. sketches and photographs) occur in different environments, with $Z$ containing this stylistic information.

In this setting, we can see that the correlation between $X$ and $Y$ would vary for different values of $e$. We argue that the correlation $Y \leftrightarrow Z \to X$ is not stable in an unseen environment $e \notin \mathcal{E}_{\text{train}}$ as it involves $E$ and we only want to learn the stable causal relation $Y \to X$. However, the learned predictor may inevitably absorb the unstable relation between $X$ and $Y$ if we simply train it on the observed train distribution $p^e(X, Y)$ with empirical risk minimization.

**Balanced Distribution.** To avoid learning the unstable relations, we propose to consider a balanced distribution:

**Definition 2.2.** *A **balanced distribution** can be written as $p^B(X, Y, Z) = p_{\mathbf{f}}(X|Y, Z)p^B(Z)p^B(Y)$, where $p^B(Y) = U\{1, 2, ..., m\}$ and $Y \perp\!\!\!\perp_B Z$.*

Here we do not specify $p^B(Z)$. Note that $p^B(X|Y, Z) = p_{\mathbf{f}}(X|Y, Z)$ is a result of the unchanged causal mechanism $Z \to X \leftarrow Y$, and that $p^B(X, Y, X) \in \mathcal{F}$ can also be regarded as constructing an new environment $B \in \mathcal{E}$. In this new distribution, $X$ and $Y$ are only correlated through the stable causal relation $Y \to X$. We want to argue that the Bayesian optimal classifier trained on such a balanced distribution would have the lowest worst-case risk, compared to Bayesian optimal classifiers trained on other environments in $\mathcal{E}$ as defined in Equation (1). To support this statement, we further assume some degree of disentanglement of the causal mechanism:

**Assumption 2.3.** *There exist functions $\mathbf{g}_Y$, $\mathbf{g}_Z$ and noise variables $\epsilon_Y$, $\epsilon_Z$, such that $(Y, Z) = \mathbf{f}^{-1}(X - \epsilon) = (\mathbf{g}_Y(X - \epsilon_Y), \mathbf{g}_Z(X - \epsilon_Z))$, and $\epsilon_Y \perp\!\!\!\perp_B \epsilon_Z$.*

The above assumption implies that $Y \perp\!\!\!\perp_B Z|X$. We can then have the following theorem[2]:

**Theorem 2.4.** *Consider a classifier $C_\psi(X) = \arg\max_Y p_\psi(Y|X)$ with parameter $\psi$. The risk of such a classifier on an environment $e \in \mathcal{E}$ is its cross entropy: $L^e(p_\psi(Y|X)) = -\mathbb{E}_{p^e(X,Y)} \log p_\psi(Y|X)$. Assume that $\mathcal{E}$ satisfies:*

$$\forall e \in \mathcal{E}, Y \not\perp\!\!\!\perp_{p^e} Z \implies \exists e' \in \mathcal{E} \ s.t. \ L^{e'}(p^e(Y|X)) - L^{e'}(p^B(Y)) > 0.$$

*Then the Bayes optimal classifier trained on any balanced distribution $p^B(X, Y)$ is **minimax optimal** across all environments in $\mathcal{E}$:*

$$p^B(Y|X) = \arg\min_{p_\psi \in \mathcal{F}} \max_{e \in \mathcal{E}} L^e(p_\psi(Y|X)).$$

The assumption implies that the environment space $\mathcal{E}$ is large and diverse enough such that a perfect classifier on one environment will always perform worse than random guessing on some other environment. Under such an assumption, no other Byes optimal classifier produced by an environment in $\mathcal{E}$ would have a better worst case OOD performance than the balanced distribution.

## 3 METHOD

We propose a two-phased method that first use a VAE to learn the underlying data distribution $p^e(X, Y, Z)$ with latent covariate $Z$ for each $e \in \mathcal{E}_{\text{train}}$, and then use the learned distribution to calculate a balancing score to create a balanced distribution based on the training data.

---

[2]See Appendix A for proofs of all theorems.

## 3.1 LATENT COVARIATE LEARNING

We argue that the underlying joint distribution of $p^e(X, Y, Z)$ can be learned and identified by a VAE, given a sufficiently large set of train environments $\mathcal{E}_{\text{train}}$. To specify the correlation between $Z$ and $Y$, we assume that the conditional distribution $p^e(Z|Y)$ is conditional factorial with an exponential family distribution:

**Assumption 3.1.** *The correlation between $Y$ and $Z$ in environment $e$ is characterized by:*

$$p^e_{\mathbf{T}, \boldsymbol{\lambda}}(Z|Y) = \prod_{i=1}^{n} \frac{Q_i(Z_i)}{W_i^e(Y)} \exp\Big[ \sum_{j=1}^{k} T_{ij}(Z_i)\lambda_{ij}^e(Y) \Big],$$

*where $Z_i$ is the $i$-th element of $Z$, $\mathbf{Q} = [Q_i]_i : \mathcal{Z} \to \mathbb{R}^n$ is the base measure, $\mathbf{W}^e = [W_i^e]_i : \mathcal{Y} \to \mathbb{R}^n$ is the normalizing constant, $\mathbf{T} = [T_{ij}]_{ij} : \mathcal{Z} \to \mathbb{R}^{nk}$ is the sufficient statistics, and $\boldsymbol{\lambda}^e = [\lambda_{ij}^e]_{ij} : \mathcal{Y} \to \mathbb{R}^{nk}$ are the $Y$ dependent parameters.*

Here $n$ is the dimension of the latent variable $Z$, and $k$ is the dimension of each sufficient statistic. Note that $k$, $\mathbf{Q}$, and $\mathbf{T}$ is determined by the type of chosen exponential family distribution thus independent of the environment. The simplified conditional factorial prior assumption is from the mean-field approximation, which can be expressed as a closed form of the true prior (Blei et al., 2017). Note that the exponential family assumption is not very restrictive as it has universal approximation capabilities (Sriperumbudur et al., 2017). We then consider the following conditional generative model in each environment $e \in \mathcal{E}_{\text{train}}$, with parameters $\theta = (\mathbf{f}, \mathbf{T}, \boldsymbol{\lambda})$:

$$p^e_\theta(X, Z|Y) = p_{\mathbf{f}}(X|Z, Y)p^e_{\mathbf{T}, \boldsymbol{\lambda}}(Z|Y). \tag{2}$$

We use a VAE to estimate the above generative model with the following evidence lower bound (ELBO) in each environment $e \in \mathcal{E}_{\text{train}}$:

$$\mathbb{E}_{\mathcal{D}^e}\left[\log p^e_\theta(X|Y)\right] \geq \mathcal{L}^e_{\theta, \phi} := \mathbb{E}_{\mathcal{D}^e}\big[ \mathbb{E}_{q^e_\phi(Z|X,Y)}\left[\log p_{\mathbf{f}}(X|Z, Y)\right] - D_{\text{KL}}(q^e_\phi(Z|X, Y)||p^e_{\mathbf{T}, \boldsymbol{\lambda}}(Z|Y)\big].$$

The KL-divergence term can be calculated analytically. To sample from the variational distribution $q^e_\phi(Z|X, Y)$, we use reparameterization trick (Kingma & Welling, 2013).

We then maximize the above ELBO $\frac{1}{|\mathcal{E}_{\text{train}}|} \sum_{e \in \mathcal{E}_{\text{train}}} \mathcal{L}^e_{\theta, \phi}$ over all training environments to obtain model parameters $(\theta, \phi)$. To show that we can uniquely recover the latent variable $Z$ up to some simple transformations, we want to show that the model parameter $\theta$ is identifiable up to some simple transformations. That is, for any $\{\theta = (\mathbf{f}, \mathbf{T}, \boldsymbol{\lambda}), \theta' = (\mathbf{f}', \mathbf{T}', \boldsymbol{\lambda}')\} \in \Theta$,

$$p^e_\theta(X|Y) = p^e_{\theta'}(X|Y), \forall e \in \mathcal{E}_{\text{train}} \implies \theta \sim \theta',$$

where $\Theta$ is the parameter space and $\sim$ represents an equivalent relation. Specifically, we consider the following equivalence relation from Motiian et al. (2017):

**Definition 3.2.** *If $(\mathbf{f}, \mathbf{T}, \boldsymbol{\lambda}) \sim_A (\mathbf{f}', \mathbf{T}', \boldsymbol{\lambda}')$, then there exists an invertible matrix $A \in \mathbb{R}^{nk \times nk}$ and a vector $\mathbf{c} \in \mathbb{R}^{nk}$, such that $\mathbf{T}(\mathbf{f}^{-1}(x)) = A\mathbf{T}'(\mathbf{f}'^{-1}(x)) + \mathbf{c}, \forall x \in \mathcal{X}$.*

When the underlying model parameter $\theta^*$ can be recovered by perfectly fitting the data distribution $p^e_{\theta^*}(X|Y)$ for all $e \in \mathcal{E}_{\text{train}}$, the joint distribution $p^e_{\theta^*}(X, Z|Y)$ is also recovered. This further implies the recovery of the prior $p^e_{\theta^*}(Z|Y)$ and the true latent variable $Z^*$. The identifiability of our proposed latent covariate learning model can then be summarized as follows:

**Theorem 3.3.** *Suppose we observe data sampled from the generative model defined according to Equation* (2), *with parameters $\theta = (\mathbf{f}, \mathbf{T}, \boldsymbol{\lambda})$. In addition to Assumption 2.1 and Assumption 3.1, we assume the following conditions holds: (1) The set $\{x \in \mathcal{X}|\phi_\epsilon(x) = 0\}$ has measure zero, where $\phi_\epsilon$ is the characteristic function of the density $p_\epsilon$. (2) The sufficient statistics $T_{ij}$ are differentiable almost everywhere, and $(T_{ij})_{1 \leq j \leq k}$ are linearly independent on any subset of $\mathcal{X}$ of measure greater than zero. (3) There exist $nk + 1$ distinct pairs $(y_0, e_0), \dots, (y_{nk}, e_{nk})$ such that the $nk \times nk$ matrix*

$$\mathbf{L} = \left( \boldsymbol{\lambda}^{e_1}(y_1) - \boldsymbol{\lambda}^{e_0}(y_0), \dots, \boldsymbol{\lambda}^{e_{nk}}(y_{nk}) - \boldsymbol{\lambda}^{e_0}(y_0) \right),$$

*is invertible. Then we have the parameters $\theta = (\mathbf{f}, \mathbf{T}, \boldsymbol{\lambda})$ are $\sim_A$-**identifiable**.*

Note that in the last assumption in Theorem 3.3, since there exists $nk + 1$ distinct points $(y_i, e_i)$, the product space $\mathcal{Y} \times \mathcal{E}_{\text{train}}$ has to be large enough. i.e. We need $m|\mathcal{E}_{\text{train}}| > nk$. The invertibility of $\mathbf{L}$ implies that $\boldsymbol{\lambda}^{e_i}(y_i) - \boldsymbol{\lambda}^{e_0}(y_0)$ need to be orthogonal to each other which further implies the diversity of environment space $\mathcal{E}$.

### 3.2 Balanced mini-batch sampling

We consider using a classic method that has been widely used in the average treatment effect (ATE) estimation — balancing score matching (Rosenbaum & Rubin, 1983) — to sample balanced mini-batches that mimic a balanced distribution shown in Figure 1b. A balancing score is used to balance the systematical difference between the treated unites and the controlled units, and to reveal the true causal effect from the observed data, which is defined as below:

**Definition 3.4.** *A **balancing score** $b(Z)$ is a function of covariate $Z$ s.t. $Z \perp\!\!\!\perp Y | b(Z)$.*

There is a wide range of functions of $Z$ that can be used as a balancing score, where the propensity score $p(Y = 1|Z)$ is the coarsest one and the covariate $Z$ itself is the finest one (Rosenbaum & Rubin, 1983). To extend this statement to non-binary treatments, we first define propensity score $s(Z)$ for $Y \in \mathcal{Y} = \{1, 2, ..., m\}$ as a vector:

**Definition 3.5.** *The **propensity score** for $Y \in \{1, 2, ..., m\}$ is $s(Z) = [p(Y = y|Z)]_{y=1}^m$.*

We then have the following theorem that applies to the vector version of propensity score $s(Z)$:

**Theorem 3.6.** *Let $b(Z)$ be a function of $Z$. Then $b(Z)$ is a balancing score, if and only if $b(Z)$ is finer than $s(Z)$. i.e. exists a function $g$ such that $s(Z) = g(b(Z))$.*

We use $b^e(Z)$ to denote the balancing score for a specific environment $e$. The propensity score would then be $s^e(Z) = [p^e(Y = y|Z)]_{y=1}^m$, which can be derived from the VAE's conditional prior $p^e_{\mathbf{T},\boldsymbol{\lambda}}(Z|Y)$ as defined in Equation (2):

$$p^e(Y = y|Z) = \frac{p^e_{\mathbf{T},\boldsymbol{\lambda}}(Z|Y = y)p^e(Y = y)}{\sum_{i=1}^m p^e_{\mathbf{T},\boldsymbol{\lambda}}(Z|Y = i)p^e(Y = i)}, \tag{3}$$

where $p^e(Y = i)$ can be directly estimated from the training data $\mathcal{D}^e$.

In practice, we adopt the propensity score computed from Equation (3) as our balancing score $(b(Z) = s^e(Z))$ and propose to construct balanced mini-batches by matching $1 \leq a \leq m - 1$ different examples with different labels but the same/closest balancing score, $b^e(Z) \in \mathcal{B}$, with each train example. The detailed sampling algorithm is shown in Algorithm 1.

---

**Algorithm 1:** Balanced Mini-batch sampling.

---

**Input:** $|\mathcal{E}_{\text{train}}|$ training datasets $\mathcal{D}^e = \{(x_i^e, y_i^e)\}_{i=1}^{N^e}$ for all $e \in \mathcal{E}_{\text{train}}$, a balancing score $b^e(z_i)$ inferred from each training data point $(x_i^e, y_i^e)$, and a distance metrics $d : \mathcal{B} \times \mathcal{B} \to \mathbb{R}$;
**Output:** A balanced batch of data $D_{balanced}$ consisting of $B \times |\mathcal{E}_{\text{train}}| \times (a + 1)$ examples;
$D_{balanced} \leftarrow$ Empty;
**for** $e \in \mathcal{E}_{train}$ **do**
    Randomly sample $B$ data points $D_{random}^e$ from $\mathcal{D}^e$;
    Add $D_{random}^e$ to $D_{balanced}$;
    **for** $(x^e, y^e) \in D_{random}^e$ **do**
        $Y_{alt} = \{y_i \sim U\{1, 2, ..., m\} \setminus \{y^e, y_1, .., y_{i-1}\} | i \in [1, a]\}$;
        Compute balancing score $b^e(z^e)$ from $(x^e, y^e)$;
        **for** $y_i \in Y_{alt}$ **do**
            $j = \arg \min_{j \in [1,N^e]} d(b^e(z_j), b^e(z^e))$ such that $y_j^e = y_i$ and $(x_j^e, y_j^e) \in \mathcal{D}^e$;
            Add $(x_j^e, y_j^e)$ to $D_{balanced}$.

---

We denote the data distribution obtained from Algorithm 1 by $\hat{p}^B(X, Y, Z, E)$, then we have:

**Theorem 3.7.** *If $d(b^e(z_j), b^e(z^e)) = 0$ in Algorithm 1, the balanced mini-batch can be regarded as sampling from a semi-balanced distribution with $\hat{p}^B(Y|Z, E) = \frac{1}{a+1}(\frac{a}{m-1} + \frac{m-a-1}{m-1}p(Y|Z, E))$. When $a = m - 1$, $\hat{p}^B(Y|Z, E) = \frac{1}{m} = p^B(Y)$.*

With perfect match at every step (i.e., $b^e(z_j) = b^e(z)$) and $a = m - 1$, we can obtain a completely balanced mini-batch sampled from the balanced distribution. However, an exact match of balancing score is unlikely in reality, so a larger $a$ will introduce more noises. This can be mitigated by choosing a smaller $a$, which on the other hand will increase the dependency between $Y$ and $Z$. So in practice, the choice of $a$ reflects a trade-off between the balancing score matching quality and the degree of dependency between $Y$ and $Z$.

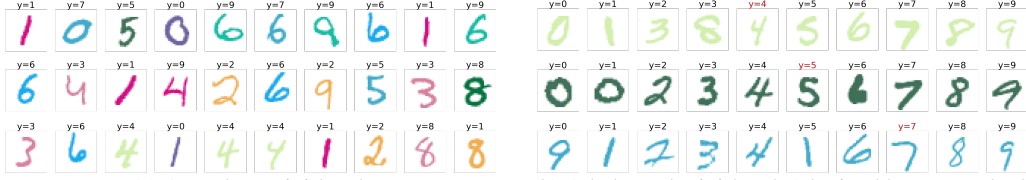

(a) A random mini-batch.      (b) A balanced mini-batch (obtained by our method).

Figure 3: A random mini-batch and a balanced mini-batch from the ColoredMNIST[10] dataset. Note that there is 25% label noise so mismatches of label $y$ and image are expected.

## 4 EXPERIMENTS

**Datasets:** To verify the effectiveness of our proposed balancing mini-batch method, we conduct experiments on DomainBed [3], a standard domain generalization benchmark, which contains seven different datasets: **ColoredMNIST** (Arjovsky et al., 2020), **RotatedMNIST** (Ghifary et al., 2015), **VLCS** (Fang et al., 2013), **PACS** (Li et al., 2017), **OfficeHome** (Venkateswara et al., 2017), **TerraIncognita** (Beery et al., 2018) and **DomainNet** (Peng et al., 2019). We also report results on a slightly modified version of **ColoredMNIST** dataset, **ColoredMNIST[10]** (Bao et al., 2021), which classify digits into 10 classes instead of binary classes.

**Baselines:** We apply our proposed balanced mini-batch sampling method along with four representative widely-used domain generalization algorithms: empirical risk minimization (**ERM**) (Vapnik, 1998), invariant risk minimization (**IRM**) (Arjovsky et al., 2020), **GroupDRO** (Sagawa et al., 2019) and deep **CORAL** (Sun & Saenko, 2016), and compare the performance of using our balanced mini-batch sampling strategy with using the usual random mini-batch sampling strategy. We compare our method with 20 baselines in total (Xu et al., 2020; Li et al., 2018a; Ganin et al., 2016; Li et al., 2018c;b; Krueger et al., 2021; Blanchard et al., 2021; Zhang et al., 2021; Nam et al., 2021; Huang et al., 2020; Shi et al., 2022; Parascandolo et al., 2021; Shahtalebi et al., 2021; Rame et al., 2022; Kim et al., 2021) reported on DomainBed, including a recent causality based baseline **CausIRL$_{CORAL}$** and **CausIRL$_{MMD}$** (Chevalley et al., 2022) that also utilize the invariance of causal mechanisms. We also compare with a group-based method **PI** (Bao et al., 2021) that interpolates the distributions of the correct predictions and the wrong predictions on ColoredMNIST[10].

To control the effect of the base algorithms, we use the same set of hyperparameters for both the random sampling baselines and our methods. We primarily consider train domain validation for model selection, as it is the most practical validation method. A detailed description of datasets and baselines, and hyperparameter tuning and selection can be found in Appendix B.

**ColoredMNIST:** We use the ColoredMNIST dataset as a proof of concept scenario, as we already know color is a dominant latent covariate that exhibits spurious correlation with the digit label.

For ColoredMNIST[10], we adopt the setting from (Bao et al., 2021), which is a multiclass version of the original ColoredMNIST dataset (Arjovsky et al., 2020). The label $y$ is assigned according to the numeric digit of the MNIST image with a 25% random noise. Then we assign one of a set of 10 colors (each indicated by a separate color channel) to the image according to the label $y$, with probability $e$ that we assign the corresponding color and probability $1 - e$ we randomly choose another color. Here $e \in \{0.1, 0.2\}$ for two train environments and $e = 0.9$ for the test environment. For ColoredMNIST, we adopt the original setting from (Arjovsky et al., 2020), which only has two classes (digit smaller/larger than 5) and two colors, with three environments $e \in \{0.1, 0.2, 0.9\}$.

**Balanced mini-batch example.** An example of a balanced mini-batch created by our method from digit 4, 5 and 7 in ColoredMNIST[10] is illustrated in Figure 3. In the random mini-batch, labels are spuriously correlated with color. e.g. most 6 are blue, most 1 are red and most 2 are yellow. In the balanced mini-batch, we force each label to have uniform color distribution by matching each example with an example with a different label but the same color. Here, the color information is implicitly learned by latent covariate learning.

**ColoredMNIST main results.** Table 1 shows the out-of-domain accuracy of our method combined with various base algorithms on ColoredMNIST[10] and ColoredMNIST dataset. Our balanced mini-

---

[3]https://github.com/facebookresearch/DomainBed

Table 1: Out-of-domain accuracy on ColoredMNIST[10] and ColoredMNIST with two train environments [0.1, 0.2] and one test environment [0.9].

| Validation | Dataset | Sampling | ERM | IRM | GroupDRO | CORAL | CausIRL | PI |
|---|---|---|---|---|---|---|---|---|
| Train | CMNIST[10] | Random | 14.25 | 13.13 | 21.06 | $13.1_{\pm 0.3}$ | $12.5_{\pm 0.1}$ | 69.68 |
| | | **Ours** | $69.8_{\pm 0.3}$ | $63.8_{\pm 0.5}$ | $69.3_{\pm 0.2}$ | $\mathbf{70.1}_{\pm 0.2}$ | $69.6_{\pm 0.3}$ | - |
| | CMNIST | Random | $10.0_{\pm 0.1}$ | $10.2_{\pm 0.3}$ | $10.0_{\pm 0.2}$ | $9.9_{\pm 0.1}$ | $10.0_{\pm 0.1}$ | - |
| | | **Ours** | $37.6_{\pm 2.9}$ | $31.1_{\pm 8.6}$ | $17.0_{\pm 3.5}$ | $\mathbf{57.2}_{\pm 3.4}$ | $43.7_{\pm 9.5}$ | - |
| Test | CMNIST[10] | Random | 26.15 | 45.41 | 32.51 | $21.1_{\pm 0.1}$ | $20.8_{\pm 0.3}$ | 69.44 |
| | | **Ours** | $\mathbf{70.5}_{\pm 0.4}$ | $63.8_{\pm 0.4}$ | $69.4_{\pm 0.3}$ | $70.1_{\pm 0.2}$ | $69.6_{\pm 0.3}$ | - |
| | CMNIST | Random | $28.7_{\pm 0.5}$ | $58.5_{\pm 3.3}$ | $36.8_{\pm 2.8}$ | $31.1_{\pm 1.6}$ | $27.4_{\pm 0.3}$ | - |
| | | **Ours** | $38.4_{\pm 3.0}$ | $\mathbf{69.7}_{\pm 16.5}$ | $44.8_{\pm 11.0}$ | $60.5_{\pm 4.1}$ | $43.3_{\pm 9.2}$ | - |

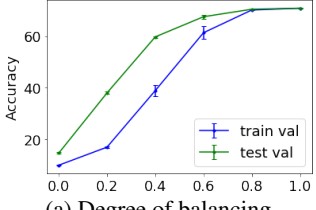
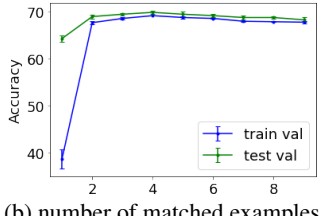
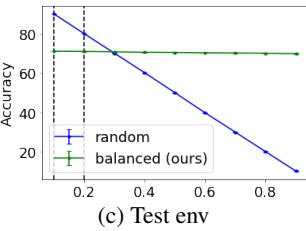

(a) Degree of balancing     (b) number of matched examples     (c) Test env

Figure 4: The out-of-domain accuracy versus (a) degree of balancing, (b) number of matched examples $a$, and (c) test environment, on ColoredMNIST[10] dataset with ERM base algorithm.

batch sampling can increase the accuracy of all base algorithms by a large margin, with CORAL improving the most (57% and 47.3%). Note that the highest possible accuracy without relying on the color feature is 75%.

In Figure 4, we study important factors in our proposed method by ablating on the ColoredMNIST[10] dataset with ERM.

**The effectiveness of balancing.** We construct oracle balanced mini-batches with $b(Z) = \text{Color}$, and then control the degree of balancing by varying the fraction of balanced examples in a mini-batch: for each randomly sampled example, with probability $\beta$, we match it with 9 examples with the same color but different labels to balance the mini-batch; otherwise, we match it with 9 examples with the same color and label to maintain the original distribution. Figure 4a shows that increasing the balancing fraction would increase the OOD performance.

**The effect of the number of matched examples $a$.** Figure 4b shows that when $a$ increases, the OOD performance first increases, then becomes stable with a slightly decreasing trend. This result is consistent with our analysis in Section 3.2, that a large $a$ will increase balancing in theory, but due to imperfection of the learning of latent covariate $Z$, large $a$ will eventually introduce more low-quality matches, which may hurt the performance. It can also be observed that we do not need a very large $a$ to reach the maximum performance.

**The effect of different test environments.** In Figure 4c, we fix the train environments as [0.1, 0.2] and test on different test environments. We report the results chosen by train domain validation, as the results with test domain validation are almost the same as the training domain validation results. The accuracy of the model trained with random mini-batches drops linearly when the test environment changes from 0.1 to 0.9, indicating that the model learns to use the color feature as the main predictive evidence. On the other hand, the accuracy of the model trained with balanced mini-batches produce by our method almost stays the same across all test domains, indicating that the model learns to use domain-invariant features.

**DomainBed:** We investigate the effectiveness of our method under different situations.

**DomainBed main results.** In Table 2, we consider combining our method with four representative base algorithms: ERM, IRM, GroupDRO, and CORAL. IRM represents a wide range of invariant representation learning baselines. GroupDRO represents group-based methods that minimize the worst group errors. CORAL represents the distribution matching algorithms that match the feature distribution across train domains. In general, our method can improve the average performance of all the base algorithms by one to two points (1.6% for ERM, IRM and GroupDRO), while CORAL

Table 2: Out-of-domain accuracy on DomainBed benchmark. Numbers are averaged over all test environments with standard deviation over 3 runs. The training domain validation scheme is used. Full results on each test environment can be found in Appendix B.4.

| Algorithm | CMNIST | RMNIST | VLCS | PACS | Office-Home | TerraInc | DomainNet | Avg |
|---|---|---|---|---|---|---|---|---|
| ERM | $51.5 \pm 0.1$ | $98.0 \pm 0.0$ | $77.5 \pm 0.4$ | $85.5 \pm 0.2$ | $66.5 \pm 0.3$ | $46.1 \pm 1.8$ | $40.9 \pm 0.1$ | 66.6 |
| IRM | $52.0 \pm 0.1$ | $97.7 \pm 0.1$ | $78.5 \pm 0.5$ | $83.5 \pm 0.8$ | $64.3 \pm 2.2$ | $47.6 \pm 0.8$ | $33.9 \pm 2.8$ | 65.4 |
| GroupDRO | $52.1 \pm 0.0$ | $98.0 \pm 0.0$ | $76.7 \pm 0.6$ | $84.4 \pm 0.8$ | $66.0 \pm 0.7$ | $43.2 \pm 1.1$ | $33.3 \pm 0.2$ | 64.8 |
| Mixup | $52.1 \pm 0.2$ | $98.0 \pm 0.1$ | $77.4 \pm 0.6$ | $84.6 \pm 0.6$ | $68.1 \pm 0.3$ | $47.9 \pm 0.8$ | $39.2 \pm 0.1$ | 66.7 |
| MLDG | $51.5 \pm 0.1$ | $97.9 \pm 0.0$ | $77.2 \pm 0.4$ | $84.9 \pm 1.0$ | $66.8 \pm 0.6$ | $47.7 \pm 0.9$ | $41.2 \pm 0.1$ | 66.7 |
| CORAL | $51.5 \pm 0.1$ | $98.0 \pm 0.1$ | $\mathbf{78.8} \pm 0.6$ | $86.2 \pm 0.3$ | $68.7 \pm 0.3$ | $47.6 \pm 1.0$ | $41.5 \pm 0.1$ | 67.5 |
| MMD | $51.5 \pm 0.2$ | $97.9 \pm 0.0$ | $77.5 \pm 0.9$ | $84.6 \pm 0.5$ | $66.3 \pm 0.1$ | $42.2 \pm 1.6$ | $23.4 \pm 9.5$ | 63.3 |
| DANN | $51.5 \pm 0.3$ | $97.8 \pm 0.1$ | $78.6 \pm 0.4$ | $83.6 \pm 0.4$ | $65.9 \pm 0.6$ | $46.7 \pm 0.5$ | $38.3 \pm 0.1$ | 66.1 |
| CDANN | $51.7 \pm 0.1$ | $97.9 \pm 0.1$ | $77.5 \pm 0.1$ | $82.6 \pm 0.9$ | $65.8 \pm 1.3$ | $45.8 \pm 1.6$ | $38.3 \pm 0.3$ | 65.6 |
| MTL | $51.4 \pm 0.1$ | $97.9 \pm 0.0$ | $77.2 \pm 0.4$ | $84.6 \pm 0.5$ | $66.4 \pm 0.5$ | $45.6 \pm 1.2$ | $40.6 \pm 0.1$ | 66.2 |
| SagNet | $51.7 \pm 0.0$ | $98.0 \pm 0.0$ | $77.8 \pm 0.5$ | $86.3 \pm 0.2$ | $68.1 \pm 0.1$ | $\mathbf{48.6} \pm 1.0$ | $40.3 \pm 0.1$ | 67.2 |
| ARM | $56.2 \pm 0.2$ | $\mathbf{98.2} \pm 0.1$ | $77.6 \pm 0.3$ | $85.1 \pm 0.4$ | $64.8 \pm 0.3$ | $45.5 \pm 0.3$ | $35.5 \pm 0.2$ | 66.1 |
| VREx | $51.8 \pm 0.1$ | $97.9 \pm 0.1$ | $78.3 \pm 0.2$ | $84.9 \pm 0.6$ | $66.4 \pm 0.6$ | $46.4 \pm 0.6$ | $33.6 \pm 2.9$ | 65.6 |
| RSC | $51.7 \pm 0.2$ | $97.6 \pm 0.1$ | $77.1 \pm 0.5$ | $85.2 \pm 0.9$ | $65.5 \pm 0.9$ | $46.6 \pm 1.0$ | $38.9 \pm 0.5$ | 66.1 |
| Fish | $51.6 \pm 0.1$ | $98.0 \pm 0.0$ | $77.8 \pm 0.3$ | $85.5 \pm 0.3$ | $68.6 \pm 0.4$ | $45.1 \pm 1.3$ | $42.7 \pm 0.2$ | 67.1 |
| Fishr | $52.0 \pm 0.2$ | $97.8 \pm 0.0$ | $77.8 \pm 0.1$ | $85.5 \pm 0.4$ | $67.8 \pm 0.1$ | $47.4 \pm 1.6$ | $41.7 \pm 0.0$ | 67.1 |
| AND-mask | $51.3 \pm 0.2$ | $97.6 \pm 0.1$ | $78.1 \pm 0.9$ | $84.4 \pm 0.9$ | $65.6 \pm 0.4$ | $44.6 \pm 0.3$ | $37.2 \pm 0.6$ | 65.5 |
| SAND-mask | $51.8 \pm 0.2$ | $97.4 \pm 0.1$ | $77.4 \pm 0.2$ | $84.6 \pm 0.9$ | $65.8 \pm 0.4$ | $42.9 \pm 1.7$ | $32.1 \pm 0.6$ | 64.6 |
| SelfReg | $52.1 \pm 0.2$ | $98.0 \pm 0.1$ | $77.8 \pm 0.9$ | $85.6 \pm 0.4$ | $67.9 \pm 0.7$ | $47.0 \pm 0.3$ | $42.8 \pm 0.0$ | 67.3 |
| CausIRL$_{CORAL}$ | $51.7 \pm 0.1$ | $97.9 \pm 0.1$ | $77.5 \pm 0.6$ | $85.8 \pm 0.1$ | $68.6 \pm 0.3$ | $47.3 \pm 0.8$ | $41.9 \pm 0.1$ | 67.3 |
| CausIRL$_{MMD}$ | $51.6 \pm 0.1$ | $97.9 \pm 0.0$ | $77.6 \pm 0.4$ | $84.0 \pm 0.8$ | $65.7 \pm 0.6$ | $46.3 \pm 0.9$ | $40.3 \pm 0.2$ | 66.2 |
| **Ours**+ERM | $60.1 \pm 1.0$ | $97.7 \pm 0.0$ | $76.1 \pm 0.3$ | $86.1 \pm 0.4$ | $67.1 \pm 0.4$ | $48.0 \pm 1.7$ | $42.6 \pm 1.0$ | 68.2 |
| **Ours**+IRM | $59.2 \pm 2.9$ | $96.8 \pm 0.1$ | $76.5 \pm 0.1$ | $85.2 \pm 0.3$ | $64.6 \pm 2.3$ | $46.5 \pm 1.2$ | $40.5 \pm 1.7$ | 67.0 |
| **Ours**+DRO | $53.9 \pm 1.3$ | $97.6 \pm 0.1$ | $76.0 \pm 0.2$ | $84.9 \pm 0.2$ | $66.5 \pm 0.5$ | $45.4 \pm 0.4$ | $40.8 \pm 0.6$ | 66.4 |
| **Ours**+CORAL | $\mathbf{66.6} \pm 1.2$ | $97.7 \pm 0.1$ | $76.4 \pm 0.5$ | $\mathbf{86.7} \pm 0.1$ | $\mathbf{69.6} \pm 0.2$ | $47.0 \pm 1.2$ | $\mathbf{43.9} \pm 0.1$ | **69.7** |

improves the most (2.2%). The reason why CORAL works the best with our method, and achieves the state-of-the-art OOD accuracy not only on average but also on ColoredMNIST, PACS, Office-Home and DomainNet dataset, is likely because our method aims to balance the data distribution and close the distribution gap between domains, which is in line with the objective of distribution matching algorithms.

Our proposed method improves the most on ColoreMNIST, OfficeHome, and DomainNet, while our method is not very effective on RotatedMNIST and VLCS.

**Reason for significant improvements.** The large improvement on ColoredMNIST (8.6% for ERM, 7.2% for IRM, 1.8% for GroupDRO and 15.1% for CORAL) is likely because the dominant latent covariate, color, is relatively easy to learn with a low dimensional VAE. The good performance on OfficeHome and DomainNet (1.7% for ERM, 6.6% for IRM, 7.5% for GroupDRO and 2.4% for CORAL) is likely because of the large number of classes. OfficeHome has 65 classes, and Domain-Net has 345 classes, while all the other datasets have less or equal to 10 classes. According to the conclusion of Theorem 3.3, a larger number of labels or environments will enable the identification of a higher dimensional latent covariate, which is more likely to capture the complex underlying data distribution.

**Reason for insignificant improvements.** The lower performance on RotatedMNIST is because the digits in each domain are all rotated by the same degree. Since classes are balanced, images in each domain are already balanced for rotation, the dominant latent covariate. As the performance with random mini-batches is already very high, the noise introduced by the matching procedure may hurt the performance. VLCS on the one hand has a pretty complex data distribution as the images from each domain are very different realistic photos collected in different ways. However, VLCS only has 5 classes and 4 domains, which only enables the identification of a very low dimensional latent covariate, which is insufficient to capture the complexity of each domain.

In practice, we suggest using our method when there is a large number of classes or domains, and preferably combined with distribution matching algorithms for domain generalization.

## 5 RELATED WORK

A growing body of work has investigated the out-of-domain (OOD) generalization problem with causal modeling. One prominent idea is to learn invariant features. When multiple training domains

are available, this can be approximated by enforcing some invariance conditions across training domains by adding a regularization term to the usual empirical risk minimization (Arjovsky et al., 2020; Krueger et al., 2021; Bellot & van der Schaar, 2020; Wald et al., 2021; Chevalley et al., 2022). There are also some group-based works (Sagawa et al., 2019; Bao et al., 2021; Liu et al., 2021b; Sanh et al., 2021; Piratla et al., 2021; Zhou et al., 2021) that improve worst group performance and can be applied to domain generalization problem. However, recent work claims that many of these approaches still fail to achieve the intended invariance property (Kamath et al., 2021; Rosenfeld et al., 2020; Guo et al., 2021), and thorough empirical study questions the true effectiveness of these domain generalization methods (Gulrajani & Lopez-Paz, 2020).

Instead of using datasets from multiple domains, Makar et al. (2022) and Puli et al. (2022) propose to utilize an additional auxiliary variable different from the label to solve the OOD problem, using a single train domain. Their methods are two-phased: (1) reweight the train data with respect to the auxiliary variable; (2) add invariance regularizations to the training objective. The limitation of such methods is that they can only handle distribution shifts induced by the chosen auxiliary variable. Little & Badawy (2019) also propose a bootstrapping method to resample train data by reweighting to mimic a randomized controlled trial. There is also single-phased methods like Wang et al. (2021) which proposes new training objectives to reduce spurious correlations.

Some other OOD works aim to improve OOD accuracy without any additional information. Liu et al. (2021a) and Lu et al. (2022) propose to use VAE to learn latent variables in the assumed causal graph, with appropriate assumptions of the train data distribution in a single train domain. The identifiability of such latent variables is usually based on Khemakhem et al. (2020), which assumes that the latent variable has a factorial exponential family distribution given an auxiliary variable. Our identifiability result is also an extension of Khemakhem et al. (2020), where we use both label $Y$ and training domain $E$ as the auxiliary variable and include the label $Y$ in the causal mechanism of generating $X$ instead of only using the latent variable $Z$ to generate $X$. Christiansen et al. (2021) use interventions on a different structural causal model to model the OOD test distributions and show a similar minimax optimal result.

To sample from the balanced distribution, we use a classic method for average treatment effect (ATE) estimation (Holland, 1986) – balancing score matching (Rosenbaum & Rubin, 1983). Causal effect estimation studies the effect a treatment would have had on a unit that in reality received another treatment. A causal graph (Pearl, 2009) similar to Figure 1a is usually considered in a causal effect estimation problem, where $Z$ is called the covariate (e.g. a patient profile), which is observed before treatment $Y \in \{0, 1\}$ (e.g. taking placebo or drug) is applied. We denote the effect of receiving a specific treatment $Y = y$ as $X_y$ (e.g. blood pressure). Note that the causal graph implies the Strong Ignorability assumption (Rubin, 1978). i.e. $Z$ includes all variables related to both $X$ and $Y$. In the case of a binary treatment, the ATE is defined as $\mathbb{E}[X_1 - X_0]$.

For a randomized controlled trial, ATE can be directly estimated by $\mathbb{E}[X|Y = 1] - \mathbb{E}[X|Y = 0]$, as in this case $Z \perp\!\!\!\perp Y$ and there would not be systematic differences between units exposed to one treatment and units exposed to another. However, in most observed datasets, $Z$ is correlated with $Y$. Thus $\mathbb{E}[X_1]$ and $\mathbb{E}[X_0]$ are not directly comparable. We can then use balancing score $b(Z)$ (Dawid, 1979) to de-correlate $Z$ and $Y$, and ATE can then be estimated by matching units with same balancing score but different treatments: $\mathbb{E}[X_1 - X_0] = \mathbb{E}_{b(Z)}[\mathbb{E}[X|Y = 1, b(Z)] - \mathbb{E}[X|Y = 0, b(Z)]]$. Recently, Schwab et al. (2018) extends this method to individual treatment effect (ITE) estimation (Holland, 1986) by constructinng virtually randomized mini-batches with balancing score.

## 6    Conclusion

Our novel causality-based domain generalization method for classification task samples balanced mini-batches to reduce the presentation of spurious correlations in the dataset. We propose a spurious-free balanced distribution and show that the Bayes optimal classifier trained on such distribution is minimax optimal over all environments. We show that our assumed data generation model with an invariant causal mechanism can be identified up to sample transformations. We demonstrate theoretically that the balanced mini-batch is approximately sampled from a spurious-free balanced distribution with the same causal mechanism under ideal scenarios. Our experiments empirically show the effectiveness of our method in both semi-synthetic settings and real-world settings.

ACKNOWLEDGMENTS

This work was supported by the National Science Foundation award #2048122. The views expressed are those of the author and do not reflect the official policy or position of the US government. We thank Google and the Robert N. Noyce Trust for their generous gift to the University of California. This work was also supported in part by the National Science Foundation Graduate Research Fellowship under Grant No. 1650114. This work was also partially supported by the National Institutes of Health (NIH) under Contract R01HL159805, by the NSF-Convergence Accelerator Track-D award #2134901, by a grant from Apple Inc., a grant from KDDI Research Inc, and generous gifts from Salesforce Inc., Microsoft Research, and Amazon Research. This work was also supported by NSERC Discovery Grant RGPIN-2022-03215, DGECR-2022-00357.

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

# A    PROOFS

In this section, we give full proofs of the main theorems in the paper.

## A.1    BALANCED DISTRIBUTION

### A.1.1    PROOF FOR THEOREM 2.4

Here we give a proof of the minimax optimality of the Bayes optimal classifier trained on a balanced distribution.

*Proof.* The Bayes optimal classifier trained on a balanced distribution $p^B(X, Y)$ has $p_\psi(Y|X) = p^B(Y|X)$. Then consider the expected cross entropy loss of such classifier on an unseen test distribution $p^e$:

$$L^e(p^B(Y|X)) = -\mathbb{E}_{p^e(X,Y)} \log p^B(Y|X) \tag{4}$$

$$= -\mathbb{E}_{p^e(X,Y)} \log p^B(Y) + \mathbb{E}_{p^e(X,Y)} \log \frac{p^B(Y)}{p^B(Y|X)}$$

$$= L^e(p^B(Y)) + \mathbb{E}_{p^e(X,Y,Z)} \left[ \log \frac{p^B(Y)}{p^B(Y|X)} \right]$$

$$= L^e(p^B(Y)) + \mathbb{E}_{p^e(Y,Z)} \left[ \mathbb{E}_{p^B(X|Y,Z)} \left[ \log \frac{p^B(Y)}{p^B(Y|X)} \right] \right]$$

$$= L^e(p^B(Y)) + \mathbb{E}_{p^e(Y,Z)} \left[ \mathbb{E}_{p^B(X|Y,Z)} \left[ \log \frac{p^B(Y|Z)}{p^B(Y|X,Z)} \right] \right] \tag{5}$$

$$= L^e(p^B(Y)) + \mathbb{E}_{p^e(Y,Z)} \left[ \mathbb{E}_{p^B(X|Y,Z)} \left[ \log \frac{p^B(X|Z)}{p^B(X|Y,Z)} \right] \right]$$

$$= L^e(p^B(Y)) - \mathbb{E}_{p^e(Y,Z)} KL[p^B(X|Y,Z)||p^B(X|Z)].$$

- Equation (4) is the definition of cross entropy loss.

- Equation (5) is obtained by $Y \perp\!\!\!\perp_B Z$ and $Y \perp\!\!\!\perp_B Z|X$.

Thus we have the cross entropy loss of $p^B(X, Y)$ in any environment $e$ is smaller than that of $p^B(Y) = \frac{1}{m}$ (random guess):

$$L^e(p^B(Y|X)) - L^e(p^B(Y)) \leq -\mathbb{E}_{p^e(Y,Z)} KL[p^B(X|Y,Z)||p^B(X|Z)] \leq 0,$$

which means:

$$\max_{e' \in \mathcal{E}} \left[ L^{e'}(p^B(Y|X)) - L^{e'}(p^B(Y)) \right] \leq 0.$$

That is, the performance of $p^B(X, Y)$ is at least as good as a random guess in any environment. Since we assume the environment diversity, that is for any $p^e$ with $Y \not\perp\!\!\!\perp_e Z$, there exists an environment $e'$ such that $p^e(Y|X)$ performs worse than a random guess. So we have:

$$\max_{e' \in \mathcal{E}} \left[ L^{e'}(p^B(Y|X)) - L^{e'}(p^B(Y)) \right] \leq 0 < \max_{e' \in \mathcal{E}} \left[ L^{e'}(p^e(Y|X)) - L^{e'}(p^B(Y)) \right].$$

Now we want to prove that $\forall e \in \mathcal{E}$, $Y \perp\!\!\!\perp_e Z$, $Y \perp\!\!\!\perp_e Z|X$, $p^e(Y) = \frac{1}{m} \implies p^e(Y|X) = p^B(Y|X)$. For any $Z \in \mathcal{Z}$, we have:

$$p^e(Y|X) = p^e(Y|X, Z)$$

$$= p^e(Y)\frac{p^e(X|Y, Z)}{\mathbb{E}_{p^e(Y|Z)}[p^e(X|Z, Y)]}$$

$$= p^B(Y)\frac{p^B(X|Y, Z)}{\mathbb{E}_{p^B(Y)}[p^B(X|Z, Y)]}$$

$$= p^B(Y|X, Z) = p^B(Y|X).$$

Thus we have the following minimax optimality:

$$p^B(Y|X) = \arg\min_{p_\psi \in \mathcal{F}} \max_{e \in \mathcal{E}} L^e(p_\psi(Y|X)).$$

$\square$

## A.2 Latent Covariate Learning

### A.2.1 Proof for Theorem 3.3

We now prove Theorem 3.3 setting up the identifiability of the necessary parameters that capture the spuriously correlated covariate features in the VAE. The proof is based on the proof of Theorem 1 in (Motiian et al., 2017), with the following modifications:

1. We use both $E$ and $Y$ as auxiliary variables.
2. We include $Y$ in the causal mechanism of generating $X$ by $X = \mathbf{f}(Y, Z) + \epsilon = \mathbf{f}_Y(Z) + \epsilon$.

*Proof.* **Step I.** In this step, we transform the equality of the marginal distributions over observed data into the equality of a noise-free distribution. Suppose we have two sets of parameters $\theta = (\mathbf{f}, \mathbf{T}, \boldsymbol{\lambda})$ and $\theta' = (\mathbf{f}', \mathbf{T}', \boldsymbol{\lambda}')$ such that $p_\theta(X|Y, E = e) = p_{\theta'}(X|Y, E = e)$, $\forall e \in \mathcal{E}_{\text{train}}$, then:

$$\int_{\mathcal{Z}} p_{\mathbf{T},\boldsymbol{\lambda}}(Z|Y, E = e)p_{\mathbf{f}}(X|Z, Y)dZ = \int_{\mathcal{Z}} P_{\mathbf{T}',\boldsymbol{\lambda}'}(Z|Y, E = e)p'_{\mathbf{f}}(X|Z, Y)dZ$$

$$\Rightarrow \quad \int_{\mathcal{Z}} p_{\mathbf{T},\boldsymbol{\lambda}}(Z|Y, E = e)p_\epsilon(X - \mathbf{f}_Y(Z))dZ = \int_{\mathcal{Z}} p_{\mathbf{T}',\boldsymbol{\lambda}'}(Z|Y, E = e)p_\epsilon(X - \mathbf{f}'_Y(Z))dZ$$

$$\Rightarrow \quad \int_{\mathcal{X}} p_{\mathbf{T},\boldsymbol{\lambda}}(\mathbf{f}^{-1}(\bar{X})|Y, E = e)\text{vol}J_{\mathbf{f}^{-1}}(\bar{X})p_\epsilon(X - \bar{X})d\bar{X} =$$

$$\int_{\mathcal{X}} p_{\mathbf{T}',\boldsymbol{\lambda}'}(\mathbf{f}'^{-1}(\bar{X})|Y, E = e)\text{vol}J_{\mathbf{f}'^{-1}(\bar{X})}p_\epsilon(X - \bar{X})d\bar{X} \quad (6)$$

$$\Rightarrow \quad \int_{\mathbb{R}^d} \tilde{p}_{\mathbf{T},\boldsymbol{\lambda},\mathbf{f},Y,e}(\bar{X})p_\epsilon(X - \bar{X})d\bar{X} = \int_{\mathbb{R}^d} \tilde{p}_{\mathbf{T}',\boldsymbol{\lambda}',\mathbf{f}',Y,e}(\bar{X}p_\epsilon(X - \bar{X})d\bar{X}) \quad (7)$$

$$\Rightarrow \quad (\tilde{p}_{\mathbf{T},\boldsymbol{\lambda},\mathbf{f},Y,e} * p_\epsilon)(X) = (\tilde{p}_{\mathbf{T}',\boldsymbol{\lambda}',\mathbf{f}',Y,e} * P_{\mathcal{E}})(X) \quad (8)$$

$$\Rightarrow \quad \mathscr{F}[\tilde{p}_{\mathbf{T},\boldsymbol{\lambda},\mathbf{f},Y,e}](\omega)\phi_\epsilon(\omega) = \mathscr{F}[\tilde{p}_{\mathbf{T}',\boldsymbol{\lambda}',\mathbf{f}',Y,e}](\omega)\phi_\epsilon(\omega) \quad (9)$$

$$\Rightarrow \quad \mathscr{F}[\tilde{p}_{\mathbf{T},\boldsymbol{\lambda},\mathbf{f},Y,e}](\omega) = \mathscr{F}[\tilde{p}_{\mathbf{T}',\boldsymbol{\lambda}',\mathbf{f}',Y,e}](\omega) \quad (10)$$

$$\Rightarrow \quad \tilde{p}_{\mathbf{T},\boldsymbol{\lambda},\mathbf{f},Y,e}(X) = \tilde{p}_{\mathbf{T}',\boldsymbol{\lambda}',\mathbf{f}',Y,e}(X). \quad (11)$$

- In Equation (6), we denote the volume of a matrix $\mathbf{A}$ as $\text{vol}\mathbf{A} := \sqrt{\det \mathbf{A}^T\mathbf{A}}$. $J$ denotes the Jacobian. We made the change of variable $\bar{X} = \mathbf{f}_Y(Z)$ on the left hand side and $\bar{X} = \bar{\mathbf{f}}_Y(Z)$ on the right hand side. Since $\mathbf{f}$ is injective, we have $\mathbf{f}^{-1}(\bar{X}) = (Y, Z)$. Here we abuse $\mathbf{f}^{-1}(\bar{X})$ to specifically denote the recovery of $Z$, i.e. $\mathbf{f}^{-1}(\bar{X}) = Z$.

- In Equation (7), we introduce

$$\tilde{p}_{\mathbf{T},\boldsymbol{\lambda},\mathbf{f},Y,e}(X) = p_{\mathbf{T},\boldsymbol{\lambda}}(\mathbf{f}_Y^{-1}(X)|Y, E = e)\text{vol}J_{\mathbf{f}_Y^{-1}}(X)\mathbb{1}_{\mathcal{X}}(X),$$

on the left hand side, and similarly on the right hand side.

- In Equation (8), we use $*$ for the convolution operator.

- In Equation (9), we use $\mathscr{F}[\cdot]$ to designate the Fourier transform. The characteristic function of $\epsilon$ is then $\phi_\epsilon = \mathscr{F}[p_\epsilon]$.

- In Equation (10), we dropped $\phi_\epsilon(\omega)$ from both sides as it is non-zero almost everywhere (by assumption (1) of the Theorem).

**Step II.** In this step, we remove all terms that are either a function of $X$ or $Y$ or $e$. By taking logarithm on both sides of Equation (11) and replacing $P_{\mathbf{T},\boldsymbol{\lambda}}$ by its expression from Equation (3) we get:

$$\log \text{vol} J_{\mathbf{f}^{-1}}(X) + \sum_{i=1}^{n}(\log Q_i(\mathbf{f}_i^{-1}(X)) - \log W_i^e(Y) + \sum_{j=1}^{k} \mathbf{T}_{i,j}(\mathbf{f}_i^{-1}(X))\lambda_{i,j}^e(Y))$$

$$= \log \text{vol} J_{\mathbf{f}'^{-1}}(X) + \sum_{i=1}^{n}(\log Q_i'(\mathbf{f}_i'^{-1}(X)) - \log W_i'^e(Y) + \sum_{j=1}^{k} \mathbf{T}_{i,j}'(\mathbf{f}_i'^{-1}(X))\lambda_{i,j}'^e(Y)).$$

Let $(e_0, y_0), (e_1, y_1), ..., (e_{nk}, y_{nk})$ be the points provided by assumption (3) of the Theorem. We evaluate the above equations at these points to obtain $k + 1$ equations, and subtract the first equation from the remaining $k$ equations to obtain:

$$\langle \mathbf{T}(\mathbf{f}^{-1}(X)), \boldsymbol{\lambda}^{e_l}(y_l) - \boldsymbol{\lambda}^{e_0}(y_0)\rangle + \sum_{i=1}^{n} \log \frac{W_i^{e_0}(y_0)}{W_i^{e_l}(y_l)}$$

$$= \langle \mathbf{T}'(\mathbf{f}^{-1}(X)), \boldsymbol{\lambda}'^{e_l}(y_l) - \boldsymbol{\lambda}'^{e_0}(y_0)\rangle + \sum_{i=1}^{n} \log \frac{W_i'^{e_0}(y_0)}{W_i'^{e_l}(y_l)}. \tag{12}$$

Let $\mathbf{L}$ be the matrix defined in assumption (3) and $\mathbf{L}'$ similarly defined for $\boldsymbol{\lambda}'$ ($\mathbf{L}'$ is not necessarily invertible). Define $b_l = \sum_{i=1}^{n} \log \frac{W_i'^{e_0}(y_0)W_i^{e_l}(y_l)}{W_i^{e_0}(y_0)W_i'^{e_l}(y_l)}$ and $\mathbf{b} = [b_l]_{l=1}^{nk}$.

Then Equation (12) can be rewritten in the matrix form:

$$\mathbf{L}^T \mathbf{T}(\mathbf{f}^{-1}(X)) = \mathbf{L}'^T \mathbf{T}'(\mathbf{f}'^{-1}(X)) + \mathbf{b}. \tag{13}$$

We multiply both sides of Equation (13) by $\mathbf{L}^{-T}$ to get:

$$\mathbf{T}(\mathbf{f}^{-1}(X)) = \mathbf{A}\mathbf{T}'(\mathbf{f}'^{-1}(X)) + \mathbf{c}. \tag{14}$$

Where $\mathbf{A} = \mathbf{L}^{-T}\mathbf{L}'$ and $\mathbf{c} = \mathbf{L}^{-T}\mathbf{b}$.

**Step III.** To complete the proof, we need to show that $\mathbf{A}$ is invertible. By definition of $\mathbf{T}$ and according to Assumption (2), its Jacobian exists and is an $nk \times n$ matrix of rank $n$. This implies that the Jacobian of $\mathbf{T}' \circ \mathbf{f}'^{-1}$ exists and is of rank $n$ and so is $\mathbf{A}$.

We distinguish two cases:

1. If $k = 1$, then $\mathbf{A}$ is invertible as $\mathbf{A} \in \mathbb{R}^{n \times n}$.

2. If $k > 1$, define $\bar{\mathbf{x}} = \mathbf{f}^{-1}(\mathbf{x})$ and $\mathbf{T}_i(\bar{x}_i) = (T_{i,1}(\bar{x}_i), ..., T_{i,k}(\bar{x}_i))$.

   Suppose for any choice of $\bar{x}_i^1, \bar{x}_i^2, ..., \bar{x}_i^k$, the family $(\frac{d\mathbf{T}_i(\bar{x}_i^1)}{d\bar{x}_i^1}, ..., \frac{d\mathbf{T}_i(\bar{x}_i^k)}{d\bar{x}_i^k})$ is never linearly independent. This means that $\mathbf{T}_i(\mathbb{R})$ is included in a subspace of $\mathbb{R}^k$ of the dimension of most $k - 1$. Let $\mathbf{h}$ be a non-zero vector that is orthogonal to $T_i(\mathbb{R})$. Then for all $x \in \mathbb{R}$, we have $\langle \frac{d\mathbf{T}_i(x)}{dx}, \mathbf{h}\rangle = 0$. By integrating we find that $\langle \mathbf{T}_i(x), \mathbf{h}\rangle = $ const.

Since this is true for all $x \in \mathbb{R}$ and a $h \neq 0$, we conclude that the distribution is not strongly exponential. So by contradiction, we conclude that there exist $k$ points $\bar{x}_i^1, \bar{x}_i^2, ... \bar{x}_i^k$ such that $(\frac{d\mathbf{T}_i(\bar{x}_i^1)}{d\bar{x}_i^1}, ..., \frac{d\mathbf{T}_i(\bar{x}_i^k)}{d\bar{x}_i^k})$ are linearly independent.

Collect these points into $k$ vectors $(\bar{\mathbf{x}}^1, ..., \bar{\mathbf{x}}^k)$ and concatenate the $k$ Jacobians $J_{\mathbf{T}}(\bar{\mathbf{x}}^l)$ evaluated at each of those vectors horizontally into the matrix $\mathbf{Q} = (J_{\mathbf{T}}(\bar{\mathbf{x}}^1), ..., J_{\mathbf{T}}(\bar{\mathbf{x}}^k))$ and similarly define $\mathbf{Q}'$ as the concatenation of the Jacobians of $\mathbf{T}'(\mathbf{f}'^{-1} \circ \mathbf{f}(\bar{\mathbf{x}}))$ evaluated at those points. Then the matrix $Q$ is invertible. By differentiating Equation (14) for each $\mathbf{x}^l$, we get:

$$\mathbf{Q} = \mathbf{A}\mathbf{Q}'.$$

The invertibility of $\mathbf{Q}$ implies the invertibility of $\mathbf{A}$ and $\mathbf{Q}'$. This completes the proof.

$\square$

### A.3 BALANCED MINI-BATCH SAMPLING

#### A.3.1 PROOF FOR THEOREM 3.6

Our proof of all possible balancing scores is an extension of the proof of Theorem 2 from (Rosenbaum & Rubin, 1983), by generalizing the binary treatment to multiple treatments.

*Proof.* First, suppose the balancing score $b(Z)$ is finer than the propensity score $s(Z)$. By the definition of a balancing score (Theorem 3.4) and Bayes' rule, we have:

$$p(Y|Z, b(Z)) = p(Y|b(Z)) \tag{15}$$

On the other hand, since $b(Z)$ is a function of $Z$, we have:

$$p(Y|Z, b(Z)) = p(Y|Z) \tag{16}$$

Equation (15) and Equation (16) give us $p(Y|b(Z)) = p(Y|Z)$. So to show $b(Z)$ is a balancing score, it is sufficient to show $p(Y|b(Z)) = p(Y|Z)$.

Let the $y$-th entry of $s(Z)$ be $s_y(Z) = p(Y = y|Z)$, then:

$$\mathbb{E}[s_y(Z)|b(Z)] = \int_{\mathcal{Z}} p(Y = y|Z = z)p(Z = z|b(Z))dz = p(Y = y|b(Z)) \tag{17}$$

But since $b(Z)$ is finer than $s(Z)$, $b(Z)$ is also finer than $s_y(Z)$, then

$$\mathbb{E}[s_y(Z)|b(Z)] = s_y(Z) \tag{18}$$

Then by Equation (17) and Equation (18) we have $P(Y = y|Z) = P(Y = y|b(Z))$ as required. So $b(Z)$ is a balancing score.

For the converse, suppose $b(Z)$ is a balancing score, but that $b(Z)$ is not finer than $s(Z)$. Then there exists $z_1$ and $z_2$ such that $s(z_1) \neq s(z_2)$, but $b(z_1) = b(z_2)$. By the definition of $s(\cdot)$, there exists $y$ such that $P(Y = y|z_1) \neq P(Y = y|z_2)$. This means, $Y$ and $Z$ are not conditionally independent given $b(Z)$, thus $b(Z)$ is not a balancing score. Therefore, to be a balancing score, $b(Z)$ must be finer than $s(Z)$.

Note that $s(Z)$ is also a balancing score, since $s(Z)$ is also a function of itself.

$\square$

### A.3.2 PROOF FOR THEOREM 3.7

We provide a proof for Theorem 3.7, demonstrating the feasibility of balanced mini-batch sampling.

*Proof.* In Algorithm 1, by uniformly sampling $a$ different labels such that $y \neq y^e$, we mean sample $Y_{\text{alt}} = \{y_1, y_2, ..., y_a\}$ by the following procedure:

$$y_1 \sim U\{1, 2, ..., m\} \setminus \{y_e\}$$
$$y_2 \sim U\{1, 2, ..., m\} \setminus \{y_e, y_1\}$$
$$\vdots$$
$$y_a \sim U\{1, 2, ..., m\} \setminus \{y_e, y_1, y_2...y_{a-1}\},$$

where $U$ denotes the uniform distribution. Suppose $D_{\text{balanced}} \sim \hat{p}^B(X, Y)$, and data distribution $\mathcal{D}^e \sim p(X, Y | E = e), \forall e \in \mathcal{E}_{\text{train}}$.

Suppose we have an exact match every time we match a balancing score, then for all $e \in \mathcal{E}_{\text{train}}$, we have

$$
\begin{aligned}
\hat{p}^B(Y|b^e(Z), E = e) =& \frac{1}{a+1} p(Y|b^e(Z), E = e) + \frac{1}{a+1}(1 - p(Y|b^e(Z), E = e)\frac{1}{m-1} + \\
&+ \frac{1}{a+1}(1 - p(Y|b^e(Z), E = e)(1 - \frac{1}{m-1})\frac{1}{m-2} + ... \\
&+ \frac{1}{a+1}(1 - p(Y|b^e(Z), E = e)(1 - \frac{1}{m-1})(1 - \frac{1}{m-2})... \\
&(1 - \frac{1}{m-a+1})\frac{1}{m-a} \\
=& \frac{1}{a+1}(\frac{a}{m-1} + \frac{m-a-1}{m-1} p(Y|b^e(Z), E = e)).
\end{aligned}
$$

By the definition of balancing score, $p(Y|Z, E = e) = p(Y|b^e(Z), E = e)$ and $\hat{p}^B(Y|Z, E = e) = \hat{p}^B(Y|b^e(Z), E = e)$, then we have

$$\hat{p}^B(Y|Z, E) = \frac{1}{a+1}(\frac{a}{m-1} + \frac{m-a-1}{m-1} p(Y|Z, E)).$$

When $a = m - 1$, we have $\hat{p}^B(Y|Z, E) = \frac{1}{m} = U\{1, 2, ..., m\}$, which means $\hat{p}^B(X, Y, Z) = p^B(X, Y, Z)$. i.e. $D_{\text{balanced}}$ can be regarded as sampled from the balanced distribution $p^B$ as defined in Definition 2.2.

$\square$

## B  EXPERIMENT DETAILS

In this section, we give more details of our experiments. We perform our experiments on the DomainBed codebase[4] (Gulrajani & Lopez-Paz, 2020).

### B.1  DATASETS

**ColoredMNIST** is a variant of the MNIST handwritten digit classification dataset. Each domain in [0.1, 0.3, 0.9] is constructed by digits spuriously correlated with their color. This dataset contains 70, 000 examples of dimensions (2, 28, 28) and 2 classes, where the class indicates if the digit is less than 5, with a 25% noise. **RotatedMNIST** is another variant of MNIST where each domain

---

[4]https://github.com/facebookresearch/DomainBed

contains digits rotated by $\alpha$ degrees, where $\alpha \in \{0, 15, 30, 45, 60, 75\}$. This dataset contains 70, 000 examples of dimensions $(1, 28, 28)$ and 10 classes, where the class indicates the digit. **PACS** comprises four domains: art, cartoons, photos, and sketches. This dataset contains 9, 991 examples of dimensions $(3, 224, 224)$ and 7 classes, where the class indicates the object in the image. **VLCS** comprises four photographic domains: Caltech101, LabelMe, SUN09, and VOC2007. This dataset contains 10, 729 examples of dimensions $(3, 224, 224)$ and 5 classes, where the class indicates the main object in the photo. **OfficeHome** includes four domains: art, clipart, product, and real. This dataset contains 15, 588 examples of dimension $(3, 224, 224)$ and 65 classes, where the class indicates the object in the image. **TerraIncognita** contains photographs of wild animals taken by camera traps at four different locations: L100, L38, L43, and L46. This dataset contains 24, 788 examples of dimensions $(3, 224, 224)$ and 10 classes, where the class indicates the animal in the image. **DomainNet** has six domains: clipart, infographics, painting, quickdraw, real, and sketch. This dataset contains 586, 575 examples of size $(3, 224, 224)$ and 345 classes.

### B.2 BASELINES

We choose **ERM**, **IRM**, **GroupDRO** and **CORAL** as base algorithms to apply our method because they are representative methods for domain generalization, and they serve as strong baselines when compared to a wide range of domain generalization methods. Empirical risk minimization (**ERM**) is a default training scheme for most machine learning problems, merging all training data into one dataset and minimizing the training errors across all training domains. Invariant risk minimization (**IRM**) represents a wide range of invariant representation learning baselines. IRM learns a data representation such that the optimal linear classifier on top of it is invariant across training domains. Group distributionally robust optimization (**GroupDRO**) represents group-based methods that minimize the worst group errors. GroupDRO performs ERM while increasing the weight of the environments with larger errors. Deep **CORAL** represents the distribution matching algorithms. CORAL matches the mean and covariance of feature distributions across training domains. According to (Gulrajani & Lopez-Paz, 2020), CORAL is the best performing domain generalization algorithm averaged across 7 datasets, compared to other 13 baselines.

### B.3 HYPERPARAMETER SELECTION

**Base algorithms**: For the architecture of image classifiers, following the DomainBed setting, we train a convolutional neural network from scratch for ColoredMNIST and RotatedMNIST datasets, and use a pre-trained ResNet50 (He et al., 2016) for all other datasets. Each experiment is repeated with 3 different random seeds. We choose the hyperparameters of base algorithms based on the default hyperparameter search with random mini-batch sampling. More specifically, we extract the hyperparameters from the official experimental logs provided in the DomainBed GitHub repository. [5] To retrieve hyperparameters, we ran the script `collect_results_detailed.py`, modified from the provided `collect_results.py` script, to collect the hyperparameters that are used to produce the DomainBed results table with train domain validation.

**Balanced mini-batch construction**: We use a multi-layer perceptron (MLP) based VAE (Kingma & Welling, 2013) to learn the latent covariate $Z$. For ColoredMNIST, ColoredMNIST[10] and RotatedMNIST, we use a 2-layer MLP with 512 neurons in each layer. For all other datasets, we use a 3-layer MLP with 1024 neurons in each layer. We choose the conditional prior $p_{\mathbf{t}}(Z|Y, E = e)$ to be a Gaussian distribution with diagonal covariance matrix. We also choose the noise distribution $p_\epsilon$ to be a Gaussian distribution with zero mean and identity variance matrix. We choose the largest possible latent dimension $n$ according to Theorem 3.3 up to 64. We choose KL divergence as our distance metric $d$ on DomainBed.

The hyperparameters we use are shown in Table 3. We control $k$ by choosing different distributions to model the latent covariate: for $k = 2$, we choose Normal distribution, and for $k = 1$, we choose Normal distribution with a fixed variance equal to the identity matrix. When choosing the latent dimension $n$, we follow the identifiability requirement $m|\mathcal{E}_{\text{train}}| > nk$ in Section 3.1, and we chose the maximum allowed $n$ up to $\lambda = 64$ for large images ($224 \times 224$) and up to $\lambda = 16$ for small images ($28 \times 28$). i.e. $n = \min\{\lfloor m|\mathcal{E}_{\text{train}}|/k \rfloor, \lambda\}$. For the distance metric $d$, we choose the KL divergence on all datasets except on ColoredMNIST[10], we choose the $L\infty$ distance. Different choice

---

[5] `https://drive.google.com/file/d/16VFQWTble6-nB5AdXBtQpQFwjEC7CChM/`

of distance metric usually does not affect the final results too much, as shown in Table 4. We tune the number of matching examples $a$ for each base algorithm with a train domain validation, and the best $a$ for each base algorithm is shown in the order of ERM/IRM/GroupDRO/CORAL in the last column of Table 3. Typically, the best $a$ for a dataset across different base algorithms is similar.

Table 3: Choice of hyperparameters for constructing balanced mini-batches, including training the VAE model for latent covariate learning ($n$, lr, batch size) and the balancing score matching ($a$, $d$).

| | $|\mathcal{E}_{\text{train}}|$ | $m$ | $k$ | $n$ | lr | batch size | $d$ | $a$ |
|---|---|---|---|---|---|---|---|---|
| ColoredMNIST[10] | 2 | 10 | 1 | 16 | 1e-3 | 64 | L$\infty$ | 4/4/4/4 |
| ColoredMNIST | 2 | 2 | 1 | 3 | 1e-3 | 64 | KLD | 1/1/1/1 |
| RotatedMNIST | 5 | 10 | 1 | 16 | 1e-3 | 64 | KLD | 1/2/1/1 |
| VLCS | 3 | 5 | 2 | 7 | 1e-4 | 32 | KLD | 2/1/1/2 |
| PACS | 3 | 7 | 2 | 10 | 1e-4 | 32 | KLD | 3/2/1/2 |
| OfficeHome | 3 | 65 | 2 | 64 | 1e-4 | 32 | KLD | 2/2/2/2 |
| TerraIncognita | 3 | 10 | 2 | 14 | 1e-4 | 32 | KLD | 2/1/1/2 |
| DomainNet | 5 | 345 | 2 | 64 | 1e-4 | 32 | KLD | 5/5/5/5 |

Table 4: Out-of-domain accuracy on ColoredMNIST[10] when using different distance metrics.

| | L1 | L2 | L$\infty$ | KLD |
|---|---|---|---|---|
| Train Val | $69.3 \pm 0.1$ | $69.5 \pm 0.1$ | $69.8 \pm 0.1$ | $69.2 \pm 0.0$ |
| Test Val | $70.2 \pm 0.5$ | $70.3 \pm 0.4$ | $70.5 \pm 0.4$ | $69.9 \pm 0.3$ |

Figure 5 shows three sets of reconstructed images with the same latent covariate $Z$ and different label $Y$ using our VAE model. We can see that $Z$ keeps the color feature and some style features, while the digit shape is changed to the closest digits belongs to class $Y$.

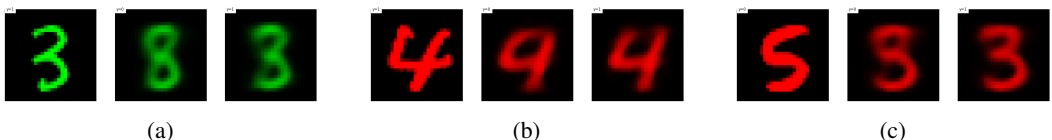

(a)        (b)        (c)

Figure 5: Reconstructed ColoredMNIST images from our VAE model. In each sub-figure, we infer $Z$ from the leftmost image, then generate images with labels $Y = 0$ (middle) and $Y = 1$ (right).

## B.4 DETAILED RESULTS

All experiments were conducted on NVidia A100, Titan RTX and RTX A6000 GPUs. Here we report detailed results on each domain of all seven datasets on DomainBed, with base algorithms ERM, IRM, GroupDRO, and CORAL. We use training domain validation.

Table 5: ColoredMNIST

| Algorithm | +90% | +80% | -90% | Avg |
|---|---|---|---|---|
| ERM | $71.7 \pm 0.1$ | $72.9 \pm 0.2$ | $10.0 \pm 0.1$ | 51.5 |
| IRM | $72.5 \pm 0.1$ | $73.3 \pm 0.5$ | $10.2 \pm 0.3$ | 52.0 |
| GroupDRO | $73.1 \pm 0.3$ | $73.2 \pm 0.2$ | $10.0 \pm 0.2$ | 52.1 |
| CORAL | $71.6 \pm 0.3$ | $73.1 \pm 0.1$ | $9.9 \pm 0.1$ | 51.5 |
| **Ours**+ERM | $71.5 \pm 0.3$ | $71.2 \pm 0.2$ | $37.6 \pm 2.9$ | 60.1 |
| **Ours**+IRM | $75.4 \pm 2.5$ | $71.0 \pm 0.3$ | $31.1 \pm 8.6$ | 59.2 |
| **Ours**+GroupDRO | $72.0 \pm 0.6$ | $72.8 \pm 0.2$ | $17.0 \pm 3.5$ | 53.9 |
| **Ours**+CORAL | $70.5 \pm 0.6$ | $72.0 \pm 0.2$ | $57.2 \pm 3.4$ | 66.6 |

Table 6: RotatedMNIST

| Algorithm | 0 | 15 | 30 | 45 | 60 | 75 | Avg |
|---|---|---|---|---|---|---|---|
| ERM | $95.9 \pm 0.1$ | $98.9 \pm 0.0$ | $98.8 \pm 0.0$ | $98.9 \pm 0.0$ | $98.9 \pm 0.0$ | $96.4 \pm 0.0$ | 98.0 |
| IRM | $95.5 \pm 0.1$ | $98.8 \pm 0.2$ | $98.7 \pm 0.1$ | $98.6 \pm 0.1$ | $98.7 \pm 0.0$ | $95.9 \pm 0.2$ | 97.7 |
| GroupDRO | $95.6 \pm 0.1$ | $98.9 \pm 0.1$ | $98.9 \pm 0.1$ | $99.0 \pm 0.0$ | $98.9 \pm 0.0$ | $96.5 \pm 0.2$ | 98.0 |
| CORAL | $95.8 \pm 0.3$ | $98.8 \pm 0.0$ | $98.9 \pm 0.0$ | $99.0 \pm 0.0$ | $98.9 \pm 0.1$ | $96.4 \pm 0.2$ | 98.0 |
| **Ours**+ERM | $94.8 \pm 0.3$ | $98.4 \pm 0.1$ | $98.7 \pm 0.0$ | $98.8 \pm 0.0$ | $98.8 \pm 0.0$ | $96.4 \pm 0.1$ | 97.7 |
| **Ours**+IRM | $93.0 \pm 0.5$ | $98.2 \pm 0.1$ | $98.6 \pm 0.1$ | $98.3 \pm 0.2$ | $98.6 \pm 0.1$ | $94.3 \pm 0.2$ | 96.8 |
| **Ours**+GroupDRO | $94.8 \pm 0.2$ | $98.5 \pm 0.1$ | $98.9 \pm 0.0$ | $98.8 \pm 0.0$ | $98.9 \pm 0.1$ | $95.9 \pm 0.3$ | 97.6 |
| **Ours**+CORAL | $94.5 \pm 0.4$ | $98.7 \pm 0.0$ | $98.8 \pm 0.1$ | $99.0 \pm 0.0$ | $98.9 \pm 0.0$ | $96.2 \pm 0.2$ | 97.7 |

Table 7: VLCS

| Algorithm | C | L | S | V | Avg |
|---|---|---|---|---|---|
| ERM | $97.7 \pm 0.4$ | $64.3 \pm 0.9$ | $73.4 \pm 0.5$ | $74.6 \pm 1.3$ | 77.5 |
| IRM | $98.6 \pm 0.1$ | $64.9 \pm 0.9$ | $73.4 \pm 0.6$ | $77.3 \pm 0.9$ | 78.5 |
| GroupDRO | $97.3 \pm 0.3$ | $63.4 \pm 0.9$ | $69.5 \pm 0.8$ | $76.7 \pm 0.7$ | 76.7 |
| CORAL | $98.3 \pm 0.1$ | $66.1 \pm 1.2$ | $73.4 \pm 0.3$ | $77.5 \pm 1.2$ | 78.8 |
| **Ours**+ERM | $96.9 \pm 0.4$ | $64.8 \pm 1.2$ | $70.2 \pm 0.8$ | $72.6 \pm 1.3$ | 76.1 |
| **Ours**+IRM | $97.5 \pm 0.3$ | $61.6 \pm 0.7$ | $72.1 \pm 1.2$ | $74.5 \pm 0.2$ | 76.5 |
| **Ours**+GroupDRO | $98.2 \pm 0.4$ | $64.0 \pm 0.9$ | $69.2 \pm 0.8$ | $72.6 \pm 0.6$ | 76.0 |
| **Ours**+CORAL | $98.3 \pm 0.1$ | $63.9 \pm 0.2$ | $69.6 \pm 1.1$ | $73.7 \pm 1.3$ | 76.4 |

Table 8: PACS

| Algorithm | A | C | P | S | Avg |
|---|---|---|---|---|---|
| ERM | $84.7 \pm 0.4$ | $80.8 \pm 0.6$ | $97.2 \pm 0.3$ | $79.3 \pm 1.0$ | 85.5 |
| IRM | $84.8 \pm 1.3$ | $76.4 \pm 1.1$ | $96.7 \pm 0.6$ | $76.1 \pm 1.0$ | 83.5 |
| GroupDRO | $83.5 \pm 0.9$ | $79.1 \pm 0.6$ | $96.7 \pm 0.3$ | $78.3 \pm 2.0$ | 84.4 |
| CORAL | $88.3 \pm 0.2$ | $80.0 \pm 0.5$ | $97.5 \pm 0.3$ | $78.8 \pm 1.3$ | 86.2 |
| **Ours**+ERM | $87.9 \pm 0.6$ | $80.5 \pm 1.0$ | $97.1 \pm 0.3$ | $79.1 \pm 1.2$ | 86.1 |
| **Ours**+IRM | $84.6 \pm 1.1$ | $79.9 \pm 0.1$ | $96.4 \pm 0.4$ | $80.0 \pm 1.2$ | 85.2 |
| **Ours**+GroupDRO | $86.3 \pm 0.6$ | $79.2 \pm 0.8$ | $96.5 \pm 0.2$ | $77.7 \pm 0.6$ | 84.9 |
| **Ours**+CORAL | $87.8 \pm 0.8$ | $81.0 \pm 0.1$ | $97.1 \pm 0.4$ | $81.1 \pm 0.8$ | 86.7 |

Table 9: OfficeHome

| Algorithm | A | C | P | R | Avg |
|---|---|---|---|---|---|
| ERM | $61.3 \pm 0.7$ | $52.4 \pm 0.3$ | $75.8 \pm 0.1$ | $76.6 \pm 0.3$ | 66.5 |
| IRM | $58.9 \pm 2.3$ | $52.2 \pm 1.6$ | $72.1 \pm 2.9$ | $74.0 \pm 2.5$ | 64.3 |
| GroupDRO | $60.4 \pm 0.7$ | $52.7 \pm 1.0$ | $75.0 \pm 0.7$ | $76.0 \pm 0.7$ | 66.0 |
| CORAL | $65.3 \pm 0.4$ | $54.4 \pm 0.5$ | $76.5 \pm 0.1$ | $78.4 \pm 0.5$ | 68.7 |
| **Ours**+ERM | $61.5 \pm 0.4$ | $53.8 \pm 0.5$ | $75.9 \pm 0.2$ | $77.4 \pm 0.5$ | 67.1 |
| **Ours**+IRM | $59.2 \pm 3.7$ | $49.8 \pm 0.9$ | $74.0 \pm 2.3$ | $75.5 \pm 2.4$ | 64.6 |
| **Ours**+GroupDRO | $61.7 \pm 1.0$ | $52.5 \pm 0.8$ | $74.9 \pm 0.8$ | $76.9 \pm 0.6$ | 66.5 |
| **Ours**+CORAL | $65.6 \pm 0.6$ | $56.5 \pm 0.6$ | $77.6 \pm 0.3$ | $78.8 \pm 0.5$ | 69.6 |

Table 10: TerraIncognita

| Algorithm | L100 | L38 | L43 | L46 | Avg |
|---|---|---|---|---|---|
| ERM | $49.8 \pm 4.4$ | $42.1 \pm 1.4$ | $56.9 \pm 1.8$ | $35.7 \pm 3.9$ | 46.1 |
| IRM | $54.6 \pm 1.3$ | $39.8 \pm 1.9$ | $56.2 \pm 1.8$ | $39.6 \pm 0.8$ | 47.6 |
| GroupDRO | $41.2 \pm 0.7$ | $38.6 \pm 2.1$ | $56.7 \pm 0.9$ | $36.4 \pm 2.1$ | 43.2 |
| CORAL | $51.6 \pm 2.4$ | $42.2 \pm 1.0$ | $57.0 \pm 1.0$ | $39.8 \pm 2.9$ | 47.6 |
| **Ours**+ERM | $53.3 \pm 0.8$ | $47.2 \pm 1.9$ | $55.3 \pm 0.7$ | $36.2 \pm 1.0$ | 48.0 |
| **Ours**+IRM | $50.0 \pm 1.9$ | $41.3 \pm 1.1$ | $54.0 \pm 2.7$ | $40.5 \pm 0.6$ | 46.5 |
| **Ours**+GroupDRO | $51.2 \pm 1.8$ | $35.4 \pm 2.5$ | $56.0 \pm 1.0$ | $38.9 \pm 1.4$ | 45.4 |
| **Ours**+CORAL | $55.2 \pm 0.3$ | $42.3 \pm 3.6$ | $54.7 \pm 0.4$ | $36.0 \pm 1.0$ | 47.0 |

Table 11: DomainNet

| Algorithm | clip | info | paint | quick | real | sketch | Avg |
|---|---|---|---|---|---|---|---|
| ERM | $58.1 \pm 0.3$ | $18.8 \pm 0.3$ | $46.7 \pm 0.3$ | $12.2 \pm 0.4$ | $59.6 \pm 0.1$ | $49.8 \pm 0.4$ | 40.9 |
| IRM | $48.5 \pm 2.8$ | $15.0 \pm 1.5$ | $38.3 \pm 4.3$ | $10.9 \pm 0.5$ | $48.2 \pm 5.2$ | $42.3 \pm 3.1$ | 33.9 |
| GroupDRO | $47.2 \pm 0.5$ | $17.5 \pm 0.4$ | $33.8 \pm 0.5$ | $9.3 \pm 0.3$ | $51.6 \pm 0.4$ | $40.1 \pm 0.6$ | 33.3 |
| CORAL | $59.2 \pm 0.1$ | $19.7 \pm 0.2$ | $46.6 \pm 0.3$ | $13.4 \pm 0.4$ | $59.8 \pm 0.2$ | $50.1 \pm 0.6$ | 41.5 |
| **Ours**+ERM | $61.2 \pm 0.2$ | $19.8 \pm 0.6$ | $48.6 \pm 0.3$ | $13.0 \pm 0.2$ | $61.0 \pm 0.4$ | $51.9 \pm 0.0$ | 42.6 |
| **Ours**+IRM | $57.9 \pm 1.6$ | $18.2 \pm 1.3$ | $46.0 \pm 1.5$ | $13.2 \pm 0.3$ | $57.2 \pm 4.5$ | $50.3 \pm 1.3$ | 40.5 |
| **Ours**+GroupDRO | $59.3 \pm 0.3$ | $18.4 \pm 0.2$ | $45.3 \pm 0.3$ | $12.2 \pm 0.4$ | $60.5 \pm 0.4$ | $48.9 \pm 0.2$ | 40.8 |
| **Ours**+CORAL | $63.4 \pm 0.1$ | $20.7 \pm 0.2$ | $50.4 \pm 0.1$ | $13.6 \pm 0.4$ | $62.7 \pm 0.1$ | $52.8 \pm 0.3$ | 43.9 |

## C  DISCUSSIONS AND LIMITATIONS

The experiments show that our balanced mini-batch sampling method outperforms the random mini-sampling baseline when applied to multiple domain generalization methods, on both semi-synthetic datasets and real-world datasets. While our method can be easily incorporated into other domain generalization methods with good performance, there are some potential drawbacks of our method. First, the computation complexity of our method grows quadratically with the dataset size, as for each training example, our method requires searching across the dataset to find the closest match in balancing score, which could become a computation bottleneck on large datasets. However, this could be solved by matching examples offline before training, or with more efficient searching methods. The second caveat is that we do not provide an optimized model selection method to complement our method. While it is possible to balance the held-out validation set with our method and choose the best model based on the accuracy of the balanced validation set, the quality of such a balanced validation set is questionable given the small size of a typical validation set. For now, we recommend the training-domain validation scheme in practice.

## D  IN-DEPTH COMPARISON WITH RELATED WORK

### D.1  COMPARISON OF ASSUMPTIONS

Certain assumptions are needed for our paper, as in other works on domain generalization. Our assumptions are not stronger than other domain generalization works that give similar generalization guarantees. Arguably, ours are weaker than most of them.

We provide the identifiability of the balanced distribution given a finite set of train environments and prove that the Bayesian optimal classifier trained on the balanced distribution would be minimax optimal across all environments. Our main assumptions are the factorial exponential distribution of the latent covariate given the label, the invertible causal function $f$, and the additive noise. Similar assumptions have been made in Sun et al. (2021).

Works without constraints on environments usually can only provide a generalization guarantee when optimizing overall environments (Mahajan et al., 2021) or do not provide any such guarantees

(Chen et al., 2021; Li et al., 2022). To provide a generalization guarantee with a single or a small number of train environments, Yuan et al. (2021); Wald et al. (2021); Ahuja et al. (2021) use a more restrictive linear causal model, Arjovsky et al. (2020) only provide full solution for linear classifiers, Christiansen et al. (2021) assume additive confounders, Yuan et al. (2021); Makar et al. (2022); Puli et al. (2022) need to utilize the observation of the variable spurious correlated with the label $Y$.

In practice, the model built with our assumptions works well on real-world datasets that do not exactly fit our assumptions, which empirically demonstrates that our method is robust against violations of our assumptions.

## D.2    COMPARISON OF CAUSAL MODEL

In general, the assumption of the underlying Structural Causal Model (SCM) is determined by the nature of the task. Sometimes, such SCM can be designed by a human expert who knows the data generation process of the task. In our paper, we propose to adopt a coarse-grained SCM for general image classification tasks with only three variables: image $X$, label $Y$, and latent variable $Z$.

Our high-level philosophy is that the image itself is merely a record of what has been done, and the label can usually be regarded as a driving force of the recorded event. When one intervenes on image $X$, the label $Y$ of the image does not necessarily change. However, if the intervention is on the class label $Y$, the image $X$ changes almost for sure for a well-defined image classification task. For example, in the medical domain, a disease ($Y$) would cause some lesions, further driving the different appearance of MRI images ($X$). Another example is when $Y$ is the object class of the item appearing in the image $X$, which is usually the case for the most widely used image classification benchmarks like ImageNet. We have also discussed this in Section 2.1 of our paper.

However, there could be exceptions. For example, if we are asked to classify whether we feel happy or sad after seeing a picture, picture $X$ would become the cause of the sentiment label $Y$. Such a scenario is less likely to happen in real-world image classification tasks. To resolve the issue of different SCM for different tasks, Christiansen et al. (2021) consider all SCMs that can be transformed into a specific linear form with plausible interventions. Wald et al. (2021) assume $X$ can be disentangled into features causing $Y$ and features caused by $Y$, and derive their theoretical results with a linear SCM. We assume a more general nonlinear SCM with $Y \rightarrow X$, which is suitable for most of the image classification tasks we consider. On the other hand, Yuan et al. (2021) directly assumes an SCM with $X \rightarrow Y$. Empirically, they obtained worse results on PACS (84.4 v.s. 86.7) and OfficeHome (64.2 v.s. 69.6) datasets, which confirms that our SCM is more suitable.

A principled way of identifying the causal relationship (if there is any) between $X$ and $Y$ is causal discovery. However, current causal discovery techniques cannot handle the complex high-dimensional image data we consider in the paper (Vowels et al., 2021). A slightly related work is Hoover (1990), which proposes that decomposing a joint distribution following the causal graph is more stable for interventions than a random decomposition. Our paper uses the invariant of $P(X|Y, Z)$, where $Z$ represents domain-dependent features like camera positions and picture style. It is hard to find such invariance in other ways of decomposition.

On the other hand, quite a few works assume no direct causal relationship between $X$ and $Y$ (Chen et al., 2021; Mahajan et al., 2021; Liu et al., 2021a; Sun et al., 2021; Ahuja et al., 2021; Li et al., 2022). Instead, they assume there is a causal feature $Z_{\text{causal}}$ directly causing $X$, together with another non-causal feature $Z_{\text{non-causal}}$. $Y$ is caused by $Z_{\text{causal}}$, which implies that $Z_{\text{causal}}$ may contain more information than $Y$. Such a causal model can be viewed as a noisy version of ours, as we consider $Y$ the same as the causal feature $Z_{\text{causal}}$, and $Z$ the same as the non-causal feature $Z_{\text{non-causal}}$. Different paper model the spurious correlation between $Z_{\text{causal}}$ and $Z_{\text{non-causal}}$ in a different way in the SCM, while we just ensure $Z_{\text{causal}}$ and $Z_{\text{non-causal}}$ are correlated, without specifying how they are correlated.

