# OpenReview forum: "Causal Balancing for Domain Generalization"
_ICLR.cc/2023/Conference — ICLR 2023 poster_

### Official Review · Reviewer_KTyH · 2022-10-24

**Confidence:** 4
**Correctness:** 4
**Technical Novelty And Significance:** 3
**Empirical Novelty And Significance:** 3
**Recommendation:** 6

**Clarity, Quality, Novelty And Reproducibility:**

This paper is well written. The proposed method is clear and the theoretical analyses is sound. The used datasets in this paper are public, and the experiments are conducted on the public codebase of DomainBed. So, I believe the experiments are fair and reproducible.

**Strength And Weaknesses:**

Strength:
1.	This paper proposes to debias the data distribution through a balanced mini-batch sampling, and then it uses the balanced data to train the model. It is technically sound.
2.	The authors give theoretical analyses for the insights of this sampling strategy.
3.	The authors implement the experiments on the public DomainBed codebase with seven datasets. It is considered fair and reproducible.
4.	The writing is good and easy to follow.
Weakness:
1.	Since there are numerous causality-based domain generalization methods, e.g., [1-9], and many of them are also based on the invariant causal structure assumption like this paper. I would like to see more in-depth discussions about the difference between them and this work. For example, some papers [1, 4] may assume that X is the cause of Y, some papers [2, 3, 5, 6, 9] may assume that there is not a direct causal relation between X and Y, but this paper supposes that Y causes X. What are the differences between them and this work? And what is the novelty of this paper in designing the causal graph in Fig. 1?
2.	As stated in Algorithm 1, the sampling strategy should be performed for each sample in the dataset. So, I guess it may not be efficient. The authors may discuss more about this limitation. For example, the total running time before/after introducing balanced sampling process.
3.	I see the main improvement of average accuracy comes from the CMNIST dataset. For example, it improves the accuracy of ERM from 51.5% to 60.1%. But the improvement on other datasets is not such significant, or even worse on RMNIST and VLCS dataset. I suggest the authors to give more discussions about this.
4.	For Fig.4 (a), if I understand correctly, training data is from the source domain and test data is from the target domain. So, I don’t see why the accuracy of training data is lower than the accuracy of test data. Could the authors explain it?
5.	Why does the results in Table 4 with KLD metrics are not consistent with the results of CMNIST in Table 2?

[1] A Causal Framework for Distribution Generalization [TPAMI 2021]
[2] A Style and Semantic Memory Mechanism for Domain Generalization [ICCV 2021]
[3] Domain Generalization using Causal Matching [ICML 2021]
[4] Learning Domain-Invariant Relationship with Instrumental Variable for Domain Generalization [arXiv 2021]
[5] Learning Causal Semantic Representation for Out-of-Distribution Prediction [NeurIPS 2021]
[6] Recovering Latent Causal Factor for Generalization to Distributional Shifts [NeurIPS 2021]
[7] On Calibration and Out-of-domain Generalization [NeurIPS 2021]
[8] Invariance Principle Meets Information Bottleneck for Out-Of-Distribution Generalization [NeurIPS 2021]
[9] Invariant Information Bottleneck for Domain Generalization [AAAI 2022]


**Summary Of The Paper:**

This paper introduces a balanced sampling strategy to make data distribution spurious-free. It is based on the causal assumption of invariant data generating process. The model trained on the unbiased data is assumed to be minimax optimal across different environments. The authors also provide identifiability guarantee of the causal model. Experiments on the DomainBed dataset show its favorable out-of-distribution generalization performance. The method is basically sound and the writing is good.

**Summary Of The Review:**

This paper introduces a causality-inspired balanced mini-batch sampling method for domain generalization. The details of the proposed method and the theoretical insights are introduced clearly. The experiments on seven popular datasets and the DomainBed codebase are fair and good. My concerns about this paper are its novelty in causal graph, its sampling efficiency, and its performance improvement.

---

> ### Author Response · Authors · 2022-11-12
> **Response**
>
> Thank you for your insightful comment. Below is our response to your review:
>
> **About more in-depth comparison to related work:** In our general response, we give a detailed discussion of why we adopt a causal graph with $Y \to X$, with a comparison to the mentioned related work. We also compare our assumptions to those made in related work in the general response. We want to respectfully point out that, the novelty of our work does not lie in the design of our causal graph, but in modeling the domain generalization in a principled causal framework and proposing a novel valid solution with useful theoretical guarantees. To the best of our knowledge, we are the first to introduce a balanced mini-batch sampling method for domain generalization.
>
> **About the efficiency of the balanced sampling process:** We match each training example with all possible counterparts offline and store this information for use at train time. So there is no sampling overhead at train time. The total offline matching time consists of the VAE inference time and the distance computing time. The first part scales linearly with the train data size, while the second scales quadratically with the train data size. In practice, the total matching time on an RTX A6000 GPU for ColoredMNIST is 5 to 10 seconds, for RotatedMNIST is around 1 minute, and for VLCS, PACS, OfficeHome, and TerraIncognita is about 5 minutes. We have also discussed such limitations of our method in Appendix C.
>
> **About the empirical results:** Yes, we observe that we got worse performance on RotatedMNIST and VLCS. We have discussed this in the second last paragraph in Section 4.2. For RotatedMNIST, as the performance is already close to 100%, the noise introduced in the matching process would likely hurt the performance. For VLCS, the number of train environments (3) and the number of classes (5) are too small to enable a sufficiently high dimensional latent to capture the complexity of each domain.
>
> **About Fig.4 (a):** ‘train val’ means we choose the model using train domain validation. ‘test val’ means we select the model using test domain validation. That is why ‘test val’ is higher than ‘train val’.
>
> **About CMNIST results:** Table 4 reports results on ColoredMNIST$^{10}$, the 10-class version of ColoredMNIST, while Table 2 reports results on ColoredMNIST, the original binary version of ColoredMNIST. That is why the numbers are different.

---

### Official Review · Reviewer_AxQ6 · 2022-10-24

**Confidence:** 3
**Clarity, Quality, Novelty And Reproducibility:** The paper is well written, and the st…
**Correctness:** 3
**Technical Novelty And Significance:** 3
**Empirical Novelty And Significance:** 4
**Recommendation:** 6

**Strength And Weaknesses:**


In general, the paper is well written, and the assumptions are well described and the theories are solid. The balancing problem is important for domain generation, and the authors have provided a valid solution to this problem.
My concerns are mainly about the assumption made in this paper. I would like to see how strong of these assumptions, for example, in assumption 3.1, whether is it possible to drop the exponential assumption. If the noise variable can not be added to the function f, how about the conclusion of this paper.
In addition, shall we have more advanced methods for addressing the balancing problem, for example representation learning and so on.

Overall, I think this is a good paper.

**Summary Of The Paper:**

This paper aims to design a domain generalization model based on causal balancing. To this end, the authors assume that the observation is causally determined by the label and some latent factors and a random variable. Based on such assumption, the authors have provided many theories to demonstrate that the causal model is identifiable. In the experiments, a lot of experiments have been conducted to demonstrate the effectiveness of the proposed model.

**Summary Of The Review:**

See the above comments.

---

> ### Author Response · Authors · 2022-11-12
> **Response**
>
> Thank you for your insightful comment. Below is our response to your review:
>
> **About our assumptions:** The exponential family assumption and the additive noise variable assumption are necessary for our identifiability result. We would like to argue that such assumptions are not very strong compared to other domain generalization works that achieve a similar generalization guarantee. For a more detailed discussion about how strong our assumptions are, we would like to refer the reviewer to our general response.
>
> Very recently, people have found that without exponential family distribution assumption, it is possible to establish the identifiability, for instance, [1]. We are in the process of understanding the alternative way to achieve a similar result. About the additive noise assumption, we hope further advances in representation learning might give more implications about whether this assumption can be relaxed.
>
> **About more advanced methods:** Yes, we believe our work can benefit from future advances in representation learning.
>
> [1] Partial Identifiability for Domain Adaptation [ICML 2022]

---

### Official Review · Reviewer_ypmL · 2022-10-26

**Confidence:** 3
**Correctness:** 3
**Technical Novelty And Significance:** 3
**Empirical Novelty And Significance:** 2
**Recommendation:** 6

**Clarity, Quality, Novelty And Reproducibility:**

The paper is reasonably well written, with some minor issues in the bibliography (Beery 2018a and 2018b are duplicates, Li 2017a and 2017b are duplicates; many citations have unusual capitalization/formatting).

**Details Of Ethics Concerns:**

No ethics concerns

**Strength And Weaknesses:**

Strengths:

The method developed in the paper is based on an interesting way to model different environments, and the idea of selecting the minimax model across the training environments is an interesting idea.

The results are very promising on ColoredMNIST and the 10-class version.

Weaknesses:

The improvements on the datasets other than ColoredMNIST are much more modest.

Some of the assumptions about how the environments are related are relatively strong (for example, it is quite possible that some of the datasets have more noise than the others).

**Summary Of The Paper:**

The paper develops a method for using multiple environments during training to obtain a model that performs well on an unseen test environment. The method has both theoretical and experimental support.

**Summary Of The Review:**

The paper develops an interesting method for attempting to generalize well to unseen test environments. The theoretical support is interesting, with some strong assumptions, and the method shows significant improvements on some datasets (with much more modest improvements on some others).

---

> ### Author Response · Authors · 2022-11-12
> **Response**
>
> Thank you for your insightful comment. Below is our response to your review:
>
> **About empirical results:** We would like to respectfully point out that while our method cannot outperform the baselines by a large margin on every dataset, all reported 20 domain generalization baselines cannot outperform the ERM baseline by more than 3% on any of the datasets in DomainBed besides ColoredMNIST. The modest improvement is more of a reflection of the hardness of the benchmark. We obtain the best performance on PACS, OfficeHome, and DomainNet across 20 baselines, while outperforming all baselines on ColoredMNIST by a large margin (>10%).
>
> **About our assumptions:** We do assume that the noise variable $\epsilon$ has the same distribution over different environments. However, for different datasets, we regard the distribution of $\epsilon$ as a hyperparameter, and one can choose different distributions for different datasets (e.g., use Gaussian distribution with different variances to reflect the noise level of a dataset). If one chooses to use a diagonal Gaussian distribution with zero mean as the noise distribution (the mean value can always be merged into $f$), then the variance would only affect the scale of the mean-squared-error (MSE) loss of the VAE, as the data log-likelihood of a Gaussian distribution is always proportional to a weighted MSE. In this case, the noise level of a dataset can also be adjusted by tuning the learning rate. We would like to refer the reviewer to our general response for a comprehensive comparison of our assumptions and the assumptions made by other domain generalization works.
>
> Thanks for pointing out the duplicated references. We have removed them in the revision.

---

### Official Review · Reviewer_2Bti · 2022-11-01

**Confidence:** 3
**Correctness:** 4
**Technical Novelty And Significance:** 3
**Empirical Novelty And Significance:** 4
**Recommendation:** 8

**Clarity, Quality, Novelty And Reproducibility:**

This paper is well presented and the results are well justified. In terms of novelty there is a slight similarity to the minimax results provided here: https://ieeexplore.ieee.org/abstract/document/9476906, I'd encourage the authors to take a look.

**Strength And Weaknesses:**

Strength: This paper provides identifiably results regarding the auto-encoder and its ability to recover the parameters of true distributions. The balanced mini-batch sampling strategy allows for the use of any ML classifier after, which is very nice. The experimental results are good.

Weakness: I would like to see a more rigorous discussion about why the DAG that models the data generating process is suitable for this problem as this would benefit readers. Additionally, the condition regarding the Cartesian product of support of Y and the support of E-train I would like to see discussed more and it's relation to dimensionality and the amount of data points across various environments needs.

**Summary Of The Paper:**

This paper aims to address OOD generalisation by providing a two step method that aims to generate samples from a 'balanced distribution' that can be used to train subsequent off the shelf classifiers that generalise to new environments well. This involves a two step procedure, the first involves learning p(V) using a Variational Autoencoder, and then using a matching method to create balanced mini batches. Finally, experimental validation is performed on DomainBed, where an improvement is demonstrated.

**Summary Of The Review:**

Overall, this paper provides a novel framework to first learn the balanced distribution that has invariance properties across different environments, and then train a classifier based on samples generated using the proposed sampling algorithm. Theorems regarding identifiability and minimax results are proved, and finally experimental validation on the DomainBed datasets show improvement in performance.

---

> ### Author Response · Authors · 2022-11-12
> **Response**
>
> Thank you for your insightful comment. Below is our response to your review:
>
> **About a rigorous discussion of DAG:** We agree that a more rigorous discussion about why we choose to use such a causal graph would benefit readers. Unfortunately, currently, there is no suitable tool to principally dissect the complex real-world image classification problem. For lower dimensional problems, causal discovery is a well-studied tool to reveal the underlying causal graph of the observed data. We also include a discussion of the reason why we choose such a causal graph in the general response.
>
> **About the Cartesian product of support of $Y$ and the support of $\mathcal{E}_{train}$:** We believe the reviewer is referring to the implication of the third assumption of Theorem 3.3. (If this is not the case, please kindly let us know). Since there have to be $nk+1$ distinct pairs of $(y_i, e_i)$, the Cartesian product of support of $Y$ and the support of $\mathcal{E}_{train}$ should be more significant than $nk$. We agree this implication about the number of label classes, the number of train domains, and the latent dimension is an important insight obtained in our paper. We have rewritten this part of our paper to make it clearer.
>
> **About the dimensionality and the number of data:** Thanks for the great suggestion. The theorems in our paper are obtained in the limit of infinite data, and they do not directly involve the dimensionality of data. We agree that it would be better to have a more principled understanding of the amount of data required. As our algorithm involves deep neural networks, it does not seem to be straightforward to have a rigorous answer to these questions. We think this is an interesting future direction for our paper.
>
> Thank you for pointing out the related paper. We found it interesting and have added it to the related work section in the revision.

---

### Author Response · Authors · 2022-11-12
**General response**

We want to thank all reviewers for their insightful feedback. Here we would like first to address two common concerns about our assumptions and our causal graph. The following arguments have also been added to Appendix D.

**Q: How strong are the assumptions?**

**A:** We agree that certain assumptions are needed for our paper, as in other works on domain generalization. At the same time, let us add that our assumptions are not stronger than other domain generalization works that give similar generalization guarantees. Arguably, ours are weaker than most of them.

We provide the identifiability of the balanced distribution given a finite set of train environments and prove that the Bayesian optimal classifier trained on the balanced distribution would be minimax optimal across all environments. Our main assumptions are the factorial exponential distribution of the latent covariate given the label, the invertible causal function $f$, and the additive noise. Similar assumptions have been made in [6].

Works without constraints on environments usually can only provide a generalization guarantee when optimizing overall environments [3,11] or do not provide any such guarantees [2,9]. To provide a generalization guarantee with a single or a small number of train environments, [11] only provide full solution for linear classifiers, [4,7,8] use a more restrictive linear causal model, [1] assume additive confounders, [4,13,14] need to utilize the observation of the variable spurious correlated with the label $Y$.

In practice, the model built with our assumptions works well on real-world datasets that do not exactly fit our assumptions, which empirically demonstrates that our method is robust against violations of our assumptions.

---

> ### Author Response · Authors · 2022-11-12
> **Continuation of general response**
>
> **Q: Why does the paper adopt such a Structural Causal Model (SCM), with $Y$ as a direct causal of $X$?**
>
> **A:** In general, the assumption of the underlying Structural Causal Model (SCM) is determined by the nature of the task. Sometimes, such SCM can be designed by a human expert who knows the data generation process of the task. In our paper, we propose to adopt a coarse-grained SCM for general image classification tasks with only three variables: image $X$, label $Y$, and latent variable $Z$.
>
> Our high-level philosophy is that the image itself is merely a record of what has been done, and the label can usually be regarded as a driving force of the recorded event. When one intervenes on image $X$, the label $Y$ of the image does not necessarily change. However, if the intervention is on the class label $Y$, the image $X$ changes almost for sure for a well-defined image classification task. For example, in the medical domain, a disease ($Y$) would cause some lesions, further driving the different appearance of MRI images ($X$). Another example is when $Y$ is the object class of the item appearing in the image $X$, which is usually the case for the most widely used image classification benchmarks like ImageNet. We have also discussed this in Section 2.1 of our paper.
>
> However, there could be exceptions. For example, if we are asked to classify whether we feel happy or sad after seeing a picture, picture $X$ would become the cause of the sentiment label $Y$. Such a scenario is less likely to happen in real-world image classification tasks. To resolve the issue of different SCM for different tasks, [1] consider all SCMs that can be transformed into a specific linear form with plausible interventions. [7] assume $X$ can be disentangled into features causing $Y$ and features caused by $Y$, and derive their theoretical results with a linear SCM. We assume a more general nonlinear SCM with $Y \to X$, which is suitable for most of the image classification tasks we consider. On the contrary, [4] directly assumes an SCM with $X \to Y$. Empirically, they obtained worse results on PACS (84.4 v.s. 86.7) and OfficeHome (64.2 v.s. 69.6) datasets than ours, which confirms that our SCM is more suitable.
>
> A principled way of identifying the causal relationship (if there is any) between $X$ and $Y$ is causal discovery. However, current causal discovery techniques cannot handle the complex high-dimensional image data we consider in the paper [10]. A slightly related work is [12], which proposes that decomposing a joint distribution following the causal graph is more stable for interventions than a random decomposition. Our paper uses the invariance of $P(X|Y,Z)$, where $Z$ represents domain-dependent features like camera positions and picture style. It is hard to find such invariance in other ways of decomposition.
>
> On the other hand, quite a few works assume no direct causal relationship between $X$ and $Y$ [2,3,5,6,8,9]. Instead, they assume there is a causal feature $Z_\text{causal}$ directly causing $X$, together with another non-causal feature $Z_\text{non-causal}$. $Y$ is caused by $Z_\text{causal}$, which implies that $Z_\text{causal}$ may contain more information than $Y$. Such a causal model can be viewed as a noisy version of ours, as we consider $Y$ the same as the causal feature $Z_\text{causal}$, and $Z$ the same as the non-causal feature $Z_\text{non-causal}$. Different paper model the spurious correlation between $Z_\text{causal}$ and $Z_\text{non-causal}$ in a different way in the SCM, while we just ensure $Z_\text{causal}$ and $Z_\text{non-causal}$ are correlated, without specifying how they are correlated.
>
> [1] A Causal Framework for Distribution Generalization [TPAMI 2021]
>
> [2] A Style and Semantic Memory Mechanism for Domain Generalization [ICCV 2021]
>
> [3] Domain Generalization using Causal Matching [ICML 2021]
>
> [4] Learning Domain-Invariant Relationship with Instrumental Variable for Domain Generalization [arXiv 2021]
>
> [5] Learning Causal Semantic Representation for Out-of-Distribution Prediction [NeurIPS 2021]
>
> [6] Recovering Latent Causal Factor for Generalization to Distributional Shifts [NeurIPS 2021]
>
> [7] On Calibration and Out-of-domain Generalization [NeurIPS 2021]
>
> [8] Invariance Principle Meets Information Bottleneck for Out-Of-Distribution Generalization [NeurIPS 2021]
>
> [9] Invariant Information Bottleneck for Domain Generalization [AAAI 2022]
>
> [10] D’ya Like DAGs? A Survey on Structure Learning and Causal Discovery [ACM Comput. Surv. 2022]
>
> [11] Invariant Risk Minimization [Arxiv 2019]
>
> [12] The logic of causal inference [Economics and Philosophy 1990]
>
> [13] Causally motivated shortcut removal using auxiliary labels [AISTATS 2022]
>
> [14] Out-of-distribution Generalization in the Presence of Nuisance-Induced Spurious Correlations [ICLR 2022]

---

### Decision · Program_Chairs · 2023-01-20

**Decision:**

Accept: poster

**Justification For Why Not Higher Score:**

The were some concerns about the strength of the assumptions and to what extent they hold for real data. In addition, while the experimental results are promising, the improvements are relatively marginal.


**Justification For Why Not Lower Score:**

OOD is a challenging problem that is gaining more attention in ML community. The paper makes a good contribution towards improving OOD by reducing spurious correlations and provides a sound theoretical study.

**Metareview: Summary, Strengths And Weaknesses:**

The paper addresses some challenges in OOD generalization by providing a data sampling strategy that gets rid of spurious correlations by exploiting invariances of the causal mechanisms of the data generation process. The paper is well-written and provides theoretical insights into the methods supported by a sound experimental study.

There were a few concerns about the assumptions made and the strength of the experiments, but those were addressed in the authors’ response.

Overall, all reviewers as well as the AC agree that it is a good contribution.


**Note From Pc:**

if the above contains the word "oral" or "spotlight" please see: "oral" presentation means -> notable-top-5% and "spotlight" means -> notable-top-25%. As stated in our emails, we are disassociating presentation type from AC recommendations